# The CIP2A-TOPBP1 complex safeguards chromosomal stability during mitosis

Mara De Marco Zompit[1], Mònica Torres Esteban[1], Clémence Mooser[1], Salomé Adam[2], Silvia Emma Rossi[2], Alain Jeanrenaud[1], Pia-Amata Leimbacher[1], Daniel Fink[1], Ann-Marie K. Shorrocks [3], Andrew N. Blackford [3], Daniel Durocher [2,4] & Manuel Stucki [1✉]

The accurate repair of DNA double-strand breaks (DSBs), highly toxic DNA lesions, is crucial for genome integrity and is tightly regulated during the cell cycle. In mitosis, cells inactivate DSB repair in favor of a tethering mechanism that stabilizes broken chromosomes until they are repaired in the subsequent cell cycle phases. How this is achieved mechanistically is not yet understood, but the adaptor protein TOPBP1 is critically implicated in this process. Here, we identify CIP2A as a TOPBP1-interacting protein that regulates TOPBP1 localization specifically in mitosis. Cells lacking CIP2A display increased radio-sensitivity, micronuclei formation and chromosomal instability. CIP2A is actively exported from the cell nucleus in interphase but, upon nuclear envelope breakdown at the onset of mitosis, gains access to chromatin where it forms a complex with MDC1 and TOPBP1 to promote TOPBP1 recruitment to sites of mitotic DSBs. Collectively, our data uncover CIP2A-TOPBP1 as a mitosis-specific genome maintenance complex.

[1] Department of Gynecology, University of Zurich and University Hospital Zurich, Schlieren, Switzerland. [2] Lunenfeld-Tanenbaum Research Institute, Mount Sinai Hospital, Toronto, ON, Canada. [3] Department of Oncology, MRC Weatherall Institute of Molecular Medicine, University of Oxford, John Radcliffe Hospital, Oxford, UK. [4] Department of Molecular Genetics, University of Toronto, Toronto, ON, Canada. ✉email: manuel.stucki@uzh.ch

Failure to accurately repair DSBs leads to genome instability, cell death or cancer[1]. Cells repair DSBs by two distinct families of pathways: non-homologous end joining (NHEJ) pathways that are the only DSB repair pathways active in G1, and homologous recombination (HR) pathways that require a sister chromatid as template and thus only occur in S and G2 phases[2]. By activating the G2/M checkpoint, cells buy time to repair most DNA breaks before entering mitosis. However, DSBs can also occur during cell division, for example as a result of under-replicated DNA regions at common fragile sites, which impede chromosome segregation and may ultimately cause DNA breakage in mitosis[3–5]. Cells rewire their response to DSBs in mitosis, at least in part to prevent telomeric fusions[6]. There is evidence that, instead of repairing them, mitotic cells stabilize chromosomal breaks until they can be safely repaired in the subsequent cell cycle phases[6–9]. How this is achieved mechanistically is not yet understood, but what is clear is that the chromatin response to DSBs is truncated in mitosis[10]. Early events such as phosphorylation of H2AX and recruitment of MDC1 are intact, but further downstream responses that in interphase regulate DSB repair pathway choice are severed by the inability of mitotic cells to recruit RNF8/RNF168, 53BP1 and BRCA1[6,8]. Instead, MDC1 mediates the recruitment of TOPBP1 to mitotic DSB sites, a process that is critical for the maintenance of chromosomal stability[11]. TOPBP1 is a highly versatile adaptor protein implicated in several aspects of genome integrity maintenance[12]. It is composed of different domains and regions, including nine BRCT domains, and an ATR activation domain (AAD; see Fig. 1a). TOPBP1 undergoes prominent enrichment at DSB sites throughout the cell cycle by at least four distinct mechanisms that involve phosphorylation-dependent interactions of alternating combinations of its four phospho-binding BRCT domains (BRCT 1, 2, 5 and 7) with the adaptor proteins 53BP1, RAD9, Treacle and MDC1[11,13–15]. TOPBP1 interacts with casein kinase 2 (CK2) phosphorylated MDC1 via its N-terminal BRCT0-2 module. This interaction is not cell cycle regulated but paradoxically, MDC1 recruits TOPBP1 by direct interaction exclusively in mitosis[11]. Here we present a possible solution to this conundrum by identifying cancerous inhibitor of protein phosphatase 2 A (CIP2A) as a TOPBP1-interacting protein that promotes TOPBP1 recruitment to sites of DSBs specifically in mitosis. We further demonstrate that loss of CIP2A leads to increased radio-sensitivity, DSB repair defects, increased micronuclei formation and chromosomal instability. Finally, we propose that CIP2A-TOPBP1 complex formation is regulated during the cell cycle by nuclear export, which spatially sequesters CIP2A from TOPBP1 in interphase cells, thus allowing their efficient interaction exclusively in mitosis.

## Results

**CIP2A is a TOPBP1 interaction partner**. We previously demonstrated that TOPBP1 is recruited to sites of DSBs in mitosis by direct phosphorylation-dependent interactions between its N-terminal BRCT domains with phosphorylated MDC1[11]. Interestingly, we observed that a C-terminal TOPBP1 deletion mutant containing BRCT domains 0-5 but lacking BRCT domains 6–8 and the AAD was recruited to sites of DSBs in G1, but not in mitosis (Fig. 1a, b). This suggested that domains/regions downstream of BRCT5 are also important for TOPBP1 DSB recruitment in mitosis. Inactivating point mutations in BRCT7 and the AAD, and deletion of BRCT6 did not affect TOPBP1 recruitment, either in interphase or in mitosis (Fig. 1b). However, deletion of a conserved region between BRCT5 and 6 (amino acids 751-899) abrogated mitotic DSB recruitment, while recruitment in interphase was unaffected (Fig. 1a, b). To test if this region constitutes a previously uncharacterized interaction surface for a protein that mediates TOPBP1 recruitment specifically in mitosis, we expressed a hemagglutinin

(HA)-tagged protein fragment corresponding to the entire region between BRCT5 and 6 (human TOPBP1 residues 740–899) in 293FT cells, followed by immunoprecipitation with HA affinity beads. Bound proteins were identified by liquid chromatography-tandem mass spectrometry (LC-MS/MS). Among the 16 proteins present only in the HA-TOPBP1(740-899) pull-down but not in the control, CIP2A (also called KIAA1524) caught our attention (Supplementary Fig. 1a, b). We previously identified this protein in a proteomic screen for interaction partners of full-length TOPBP1[15]. Moreover, it was also identified in a proximity proteomic screen for BRCA1-interacting factors along with TOPBP1[16] and it was recently shown to interact with TOPBP1 by yeast-two-hybrid assays and co-immonoprecipitation[17,18]. CIP2A is a 905 amino acid protein, roughly composed of two structurally distinct regions: a N-terminal Armadillo repeat domain (ArmRD; amino acids 1–560) and a C-terminal predicted coiled-coil region (Supplementary Fig. 1c)[19]. CIP2A was originally described as an endogenous inhibitory factor of protein phosphatase 2a (PP2A), and the protein is overexpressed in multiple cancers[20,21]. Interestingly, genotoxin sensitivity profiling by CRISPR-Cas9 drop-out screens revealed high sensitivity of CIP2A loss to ATR inhibitors and drugs that induce DSBs in proliferating cells such as topoisomerase I and II inhibitors[22,23]. Inspection of one of these DNA damage chemogenetic datasets[23] revealed a significant drug sensitivity correlation between MDC1 and CIP2A (correlation coefficient 0.76; Supplementary Fig. 1d). Combined, these were cues to further investigate a potential link between CIP2A and TOPBP1 in the mitotic DSB response. Western blotting confirmed that CIP2A was specifically pulled down by the conserved region between BRCT5 and 6 of TOPBP1 (Fig. 1c). Moreover, this region of TOPBP1 also pulled down the bacterially expressed and purified N-terminal CIP2A ArmRD, indicating that the interaction is direct and occurs within the ArmRD (Fig. 1d). Co-immunoprecipitation with overexpressed Flag-tagged full-length CIP2A did not show an increased interaction with endogenous TOPBP1 in Nocodazole-arrested mitotic cells and was independent of DSB induction by ionizing radiation (IR; Fig. 1e, f). Together, these findings establish CIP2A as a constitutive TOPBP1 interacting factor that binds to a conserved sequence region between BRCT5 and 6 in TOPBP1.

**CIP2A interacts with TOPBP1 at sites of DSBs in mitosis**. Consistent with a previous report[24] we found that in undamaged cells, CIP2A is localizing to the cytoplasm in interphase and accumulates on centrosomes along with TOPBP1 in mitosis (Fig. 2a, Supplementary Fig. 2a). Strikingly, while it was not detectable at sites of DSBs in interphase cells, CIP2A was markedly enriched in foci on condensed mitotic chromosomes upon irradiation, where it co-localized with TOPBP1 and the DSB marker γH2AX, but not with centromeric regions. (Fig. 2b–d, Supplementary Fig. 2a–c). CIP2A was also recruited to DSBs generated by a mCherry-LacI-Fok1 nuclease fusion protein, which induces DSBs within a single genomic locus in U2OS cells (U2OS-DSB reporter[25,26]). CIP2A recruitment occurred to Fok1-induced DSBs specifically in mitosis, but not in interphase cells (Supplementary Fig. 2d). By using high-resolution microscopy, we had previously observed that many IR-induced TOPBP1 structures in mitosis assumed the shape of filamentous assemblies[11]. Airyscan confocal microscopy showed that CIP2A was also present in such filaments on mitotic chromosomes and consistently co-localized with TOPBP1 (Fig. 2e). To test if CIP2A and TOPBP1 interact at sites of DSBs in mitosis we performed proximity ligation assays (in situ PLA). In non-irradiated mitotic cells, the PLA signal was evenly distributed throughout the cell and occasionally accumulated in a few large foci in a subset of cells. Upon irradiation, the PLA signal was not increased, but mostly concentrated in many foci that colocalized with DAPI staining, indicating that CIP2A and

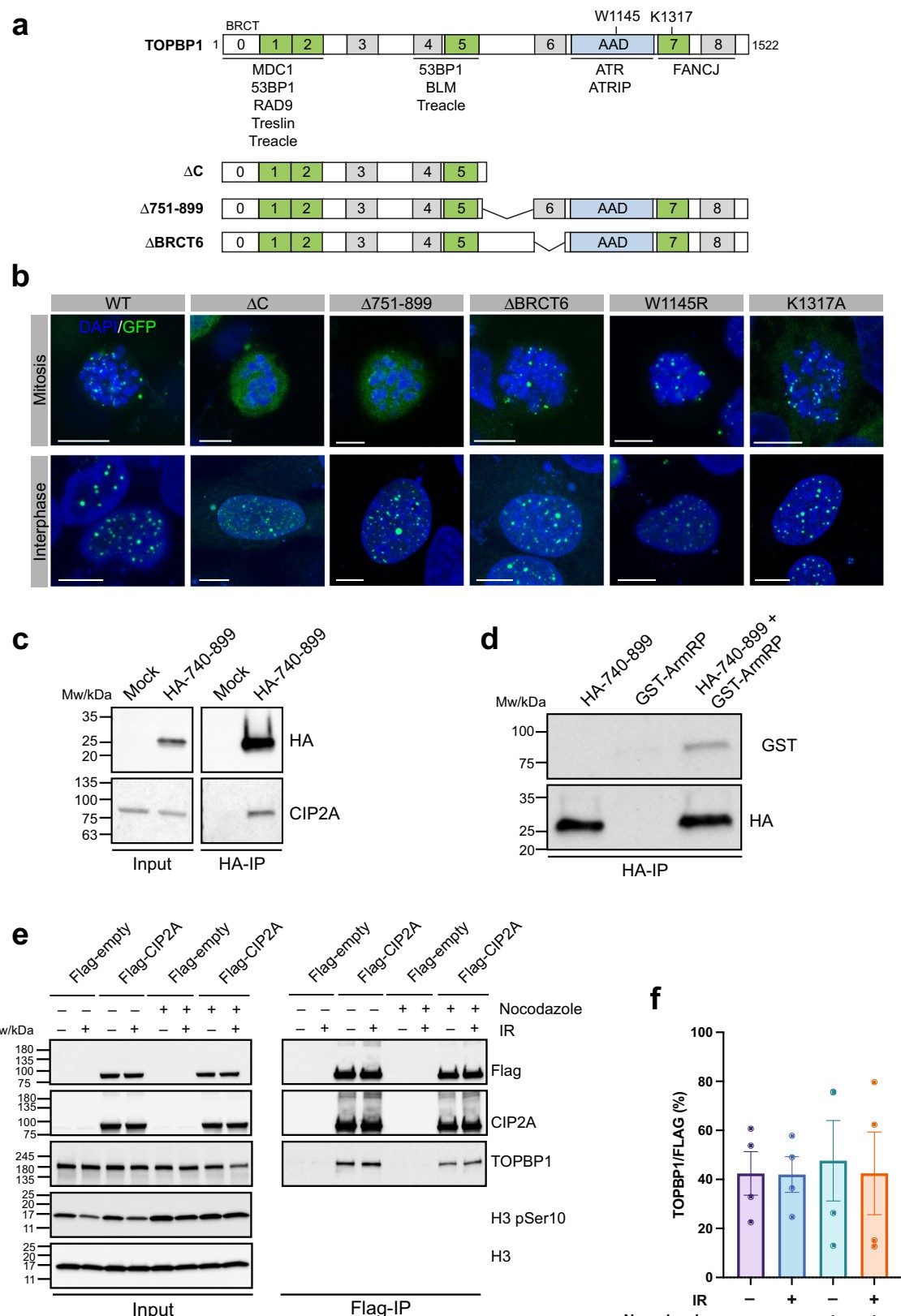

TOPBP1 indeed interact at sites of DSBs on mitotic chromosomes (Fig. 2f and Supplementary Fig. 2e). Importantly, the PLA signal was strongly reduced by depletion of TOPBP1, indicating that it is dependent on the presence of both interaction partners (Fig. 2f). These data suggest that CIP2A is recruited to sites of DSBs exclusively in mitosis where it co-localizes with TOPBP1.

**TOPBP1 accumulation at sites of mitotic DSBs is dependent on CIP2A and vice versa.** Next, we sought to determine if CIP2A is required for TOPBP1 recruitment to DSBs on mitotic chromosomes. To this end, we knocked out endogenous CIP2A in the hTERT immortalized non-transformed human retinal pigmented epithelial cell line RPE-1, using CRIPSR/Cas9 (henceforth termed RPE-1

**Fig. 1 CIP2A is a TOPBP1 interaction partner. a** Schematic showing the layout of conserved domains and regions in TOPBP1. Key amino acids in the AAD (W1145) and BRCT7 (K1317) are indicated. Deletion constructs of TOPBP1 used in **b**, lacking either the C-terminal portion of the protein (ΔC), the region between BRCT5 and 6 (Δ751-899) and BRCT domain 6 (ΔBRCT6). **b** Localization of GFP-TOPBP1 wild type and mutants in mitotic cells after 1 Gy of IR and interphase cells after 3 Gy of IR. Displayed are maximum intensity projections of confocal z-stacks. All scalebars = 10 μm. **c** HA-immunoprecipitation from 293FT cells transfected with a HA-tagged TOPBP1 fragment spanning the entire region between BRCT5 and 6 (amino acids 740-899). **d** HA-immuno-precipitation of the TOPBP1 fragment in **c** with the purified GST-ArmRP of CIP2A. **e** Flag-immunoprecipitation from 293FT transfected with Flag-tagged full-length CIP2A and treated with 3 Gy of IR and Nocodazole as indicated. **f** Quantification of the experiment in **e**. Columns represent the mean ratio between TOPBP1 and Flag band intensities in the Flag-IP blots, error bars represent the SEM of four independent experiments. Source data are provided as a Source Data file.

ΔCIP2A; Fig. 3a and Supplementary Fig. 3a, b). TOPBP1 foci were completely absent in irradiated mitotic RPE-1 ΔCIP2A cells, but they were fully restored by stable transduction of RPE-1 ΔCIP2A cells with a lentiviral vector containing Flag-tagged wild type CIP2A cDNA (Fig. 3b, c), thus ruling out off-target effects of the guide RNA. TOPBP1 enrichment at mitotic DSBs was also completely defective in U2OS cells in which CIP2A was depleted by siRNA transfection, indicating that this effect is not cell type specific (Supplementary Fig. 3c). Depletion of TOPBP1 by two different siRNAs also abrogated CIP2A foci in U2OS cells and in RPE-1 ΔCIP2A cells stably expressing Flag-tagged wild type CIP2A (Fig. 3d–f and Supplementary Fig. 3d, e). Notably, depletion of TOPBP1, but not depletion of CIP2A increased the γH2AX signal in irradiated mitotic cells, indicating that loss of CIP2A is not associated with an increased number of mitotic DSBs (Supplementary Fig. 3f). These data suggest that for efficient accumulation at sites of mitotic DSBs, TOPBP1 and CIP2A are dependent upon each other.

**Two conserved sequence segments in TOPBP1 mediate its interaction with CIP2A**. To test if interaction with CIP2A is required for TOPBP1 recruitment to sites of mitotic DNA breaks, we first sought to map the CIP2A interaction site in TOPBP1 more precisely. The region between BRCT5 and 6 in TOPBP1 contains two highly conserved amino acid segments: a short 18 amino acid patch between amino acids 778-796 and a longer conserved region between amino acids 813-892 (Supplementary Fig. 4a). Co-immunoprecipitation experiments with GFP-tagged full-length TOPBP1 and deletion mutants showed completely defective interaction with Flag-tagged full-length CIP2A, when either the entire region between BRCT5 and 6 was deleted, or when the two conserved sequence patches were deleted in combination (Fig. 4a, Supplementary Fig. 4b). Individual deletion of the short 18 amino acid patch between amino acids 774-798 also abrogated the interaction, while deletion of the longer patch between amino acids 813-892 only significantly reduced the interaction when GFP-TOPBP1 was used as the bait, but not in the reverse co-immunoprecipitation with Flag-CIP2A as the bait (Fig. 4a, Supplementary Fig. 4b). To test if TOPBP1 must exist in a complex with CIP2A in order to be recruited to sites of mitotic chromosome breaks, we knocked-down endogenous TOPBP1 expression in U2OS cells by 3'-UTR-targeting siRNA followed by re-expression of GFP-tagged wild type TOPBP1 and the deletion mutants described above. CIP2A efficiently accumulated in foci only in the presence of wild type TOPBP1, but not in the presence of deletion mutants, thus indicating that the efficient recruitment of both of these proteins to sites of mitotic DNA breaks is dependent on their interaction (Fig. 4b, c, Supplementary Fig. 4c). We conclude from these results that CIP2A and TOPBP1 are recruited to sites of DSBs in mitosis by binding to each other in a manner that is dependent on the conserved region between BRCT5 and 6 in TOPBP1.

**CIP2A-TOPBP1 recruitment to sites of mitotic DSBs is mediated by MDC1**. We next investigated the role of MDC1 in the recruitment of CIP2A and TOPBP1 to sites of mitotic DSBs.

We previously demonstrated that in mitosis, TOPBP1 recruitment is dependent on a direct interaction between its N-terminal BRCT1 and BRCT2 domains with the two conserved phosphorylated Serine residues S168 and S196, respectively[11]. CIP2A and TOPBP1 recruitment were significantly reduced, but not completely abrogated, in RPE-1 MDC1 knock-out cells (RPE-1 ΔMDC1) and in RPE-1 H2AX$^{S139A/S139A}$ knock-in cells, as well as in U2OS MDC1 knock-out cells (U2OS ΔMDC1), indicating that the majority of IR-induced CIP2A/TOPBP1 structures on mitotic chromosomes are dependent on a γH2AX-MDC1 mediated recruitment mechanism (Fig. 5a, b and Supplementary Fig. 5a, b). Consistent with the notion that CIP2A acts downstream of γH2AX and MDC1 in the mitotic response to DSBs, MDC1 recruitment is not affected in RPE-1 ΔCIP2A cells (Supplementary Fig. 5c). To test if CIP2A exists in a complex with MDC1, we co-transfected 293FT cells with Flag-tagged CIP2A and HA-tagged MDC1 (WT and S168A/S196A double mutant), followed by immunoprecipitation with anti-Flag affinity beads. Both TOPBP1 and MDC1 co-immunoprecipitated with Flag-CIP2A (Fig. 5c). Interestingly, while the interaction between CIP2A and TOPBP1 again occurred independently of DNA damage (see also Fig. 1e), the interaction between CIP2A and MDC1 was enhanced upon IR treatment and reduced when Ser168 and Ser196 in MDC1 were mutated to Ala. This indicates that CIP2A and TOPBP1 can exist in a complex with MDC1, and that formation of this ternary complex is stimulated by IR in vitro (Fig. 5c). To confirm that CIP2A is in close proximity to MDC1 also in vivo we performed in situ PLA with antibodies against CIP2A and MDC1. A PLA signal was detectable in mitotic cells and this signal was dependent on MDC1 and was enriched in foci upon irradiation (Fig. 5d and Supplementary Fig. 5d). In agreement with the in vitro interaction results, the CIP2A-MDC1 PLA signal was increased upon treatment of cells with IR, while the CIP2A-TOPBP1 PLA signal was not (compare Fig. 2f and Fig. 5d). We also observed that both CIP2A and TOPBP1 foci were absent from irradiated mitotic chromosomes in U2OS ΔMDC1 cells stably expressing GFP-tagged S168A/S196A MDC1, while they were readily detectable and co-localized with MDC1 in wild type GFP-MDC1 expressing cells (Fig. 5e, f). This indicates that accumulation of both CIP2A and TOPBP1 at sites of mitotic DSBs is dependent on the direct interaction between TOPBP1 BRCT1 and BRCT2 with phosphorylated MDC1.

**CIP2A controls TOPBP1 recruitment independently of PP2A**. Since CIP2A was described as an endogenous inhibitor of PP2A, we considered the possibility that CIP2A may promote TOPBP1 foci formation in mitosis by preventing PP2A-mediated MDC1 de-phosphorylation at Ser168 and Ser196. However, MDC1 phosphorylation at Ser168 and S196 was not affected by loss of CIP2A (Fig. 6a). Moreover, in CIP2A depleted cells, TOPBP1 interacted as efficiently with MDC1 as in CIP2A expressing control cells, indicating that CIP2A does not simply promote the association of TOPBP1 with MDC1 by preventing its de-phosphorylation (Fig. 6b). Consistent with this interpretation, treatment of RPE-1 ΔCIP2A cells with the PP2A inhibitor LB-100 was unable to rescue TOPBP1 foci in mitosis (Supplementary

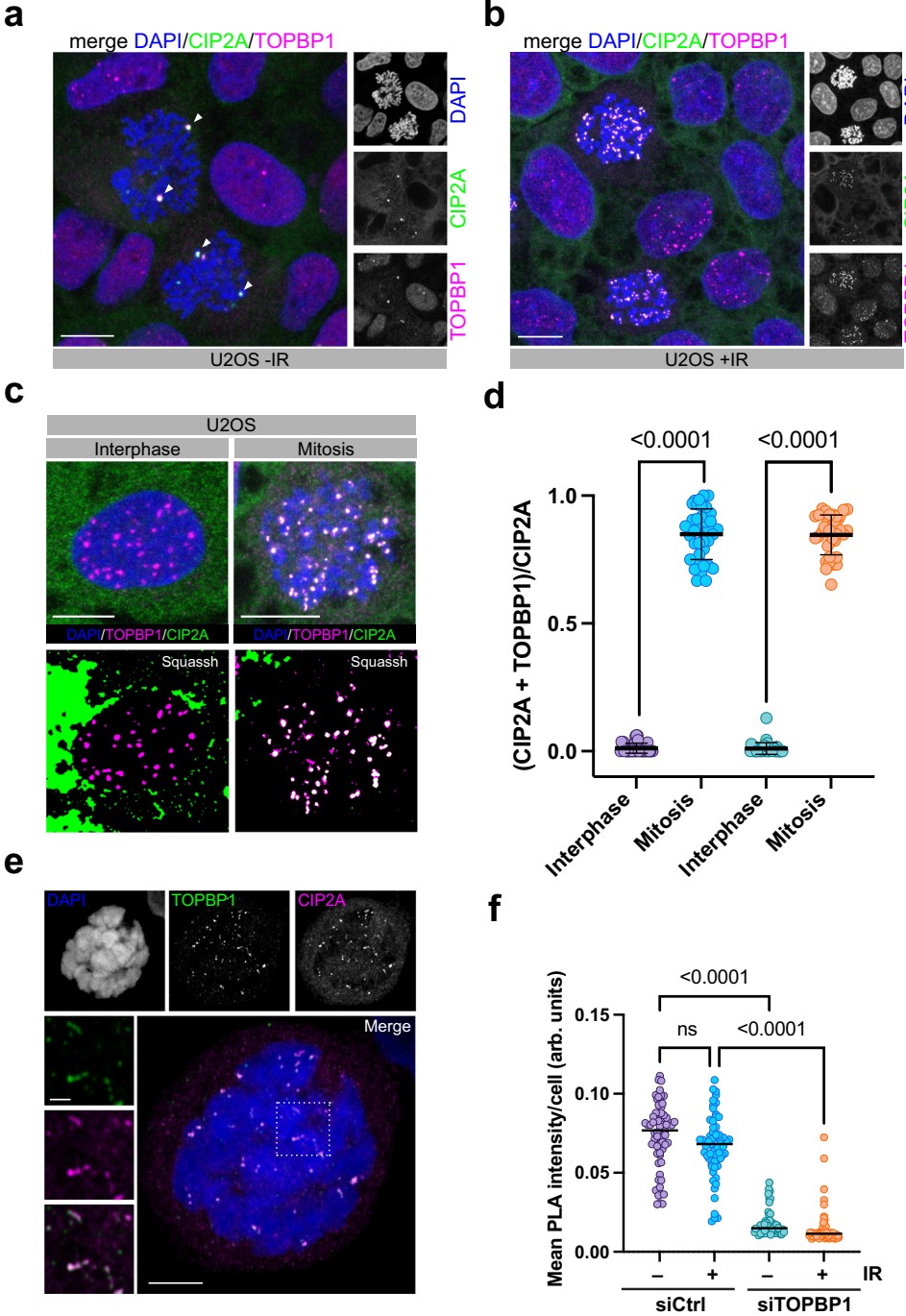

**Fig. 2 CIP2A interacts with TOPBP1 at sites of DSBs in mitosis. a** Confocal micrograph (maximum intensity projection) of untreated U2OS cells, stained for TOPBP1 and CIP2A. Centrosomes are highlighted with white arrowheads. **b** Confocal micrograph (maximum intensity projection) of Nocodazole-arrested U2OS cells 1 h after irradiation with 1 Gy, stained for TOPBP1 and CIP2A. **c** Upper panels: confocal micrographs of interphase U2OS cells treated with 3 Gy and U2OS cells arrested in mitosis by Nocodazole and treated with 1 Gy. Lower panels: micrographs deconvoluted and segmented by SQUASSH **d** Quantitative analysis of CIP2A and TOPBP1 colocalization by SQUASSH. Left: object size colocalization (area of object overlap divided by total object area). Right: object number colocalization (fraction of objects in each channel that overlap ≥ 50%). Each data point represents one cell (*n* = 37; pooled from three independent experiments). Bars and error bars represent mean and SD. Statistical significance was assessed by two-sided unpaired t-tests (α = 0.05) **e** Airyscan high-resolution confocal image (maximum intensity projection) of CIP2A and TOPBP1 foci in mitosis 1 h after 1 Gy of IR. Scale bar in the merge panel: 5 μm; scale bar in the zoomed panels: 1 μm. **f** Quantification of CIP2A-TOPBP1 proximity by in situ PLA in U2OS cells transfected with either control siRNA (siCtrl) or TOPBP1 siRNA (siTOPBP1), arrested in mitosis by Nocodazole and mock treated or treated with 1 Gy of IR. Each data point represents one cell (*n* = 61; pooled from two independent experiments), and bars represent median. Statistical significance was assessed by Kruskal-Wallis test and Dunn's multiple comparison test (α = 0.05; ns = not significant). All scale bars = 10 μm unless indicated otherwise. Source data are provided as a Source Data file.

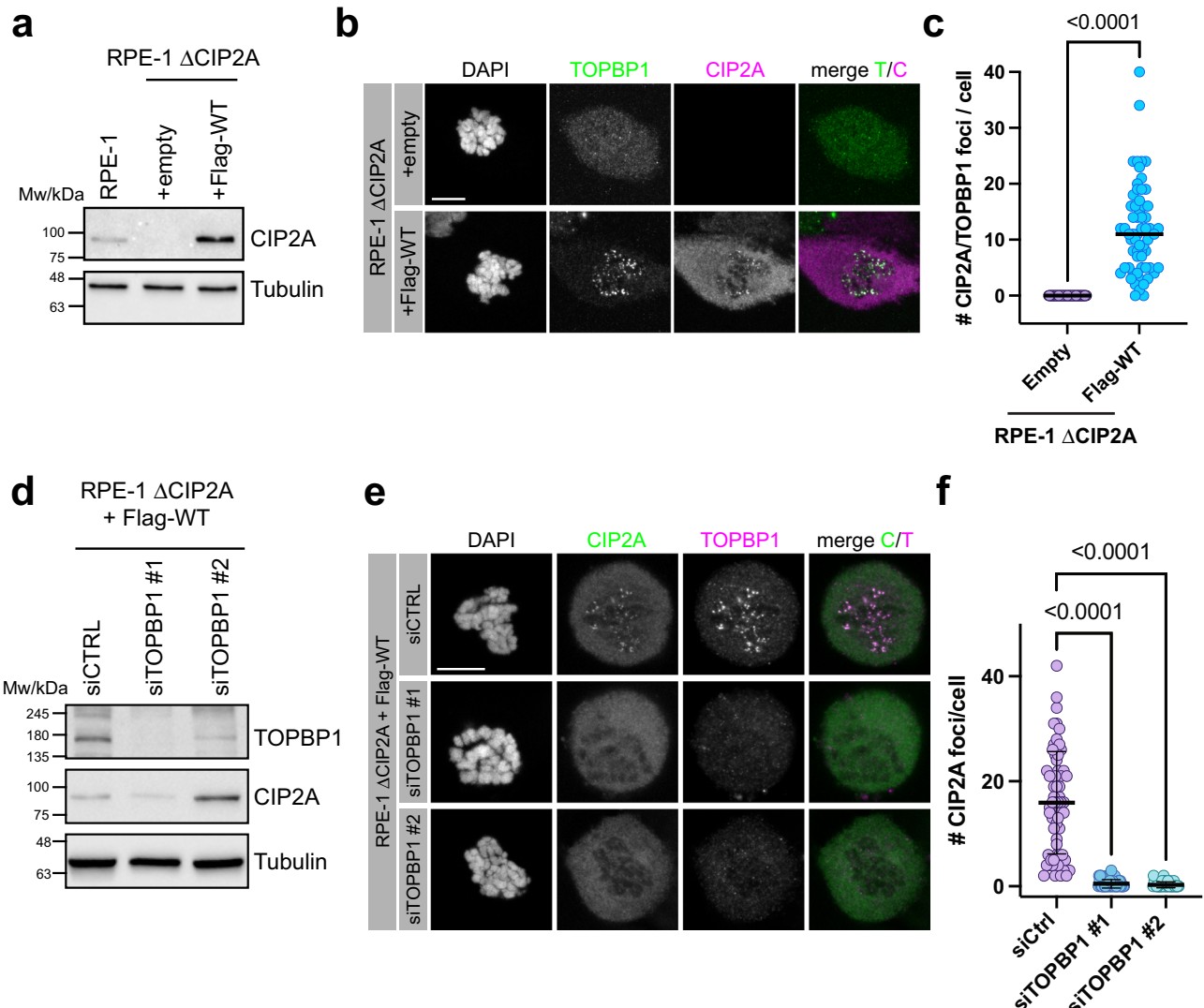

**Fig. 3 TOPBP1 accumulation at sites of mitotic DSBs is dependent on CIP2A and vice versa. a** Western blots of total cell extract of parental RPE-1 cells, RPE-1 ΔCIP2A cells and stably transduced RPE-1 ΔCIP2A cells with empty vector (+empty) and vector containing Flag-tagged wild type CIP2A cDNA (+Flag-WT). **b** Confocal micrographs (maximum intensity projections) of Nocodazole-arrested empty vector (+Empty) and Flag-tagged CIP2A wild type (+Flag-WT) complemented RPE-1 ΔCIP2A cells, treated with 1 Gy of IR and stained for TOPBP1 and CIP2A. **c** Quantification of the experiment in **b**. Number of CIP2A/TOPBP1 foci per cell was assessed (+Empty: $n = 57$, +Flag-WT: $n = 62$, pooled from three independent experiments). Statistical significance was assessed with the two-sided Mann-Whitney test ($\alpha = 0.05$) **d** Western blots of total cell extracts of CIP2A knock-out RPE-1 cells stably transduced with Flag-tagged wild type CIP2A cDNA (RPE-1 ΔCIP2A + Flag-WT) and transiently transfected with control siRNA (siCtrl) and two different siRNAs against TOPBP1 (siTOPBP1 #1 and siTOPBP1 #2). **e** Confocal micrographs (maximum intensity projection) of CIP2A knock-out RPE-1 cells stably transduced with Flag-tagged wild type CIP2A cDNA (RPE-1 ΔCIP2A + Flag-WT) and transiently transfected with control siRNA (siCtrl) and two different siRNAs against TOPBP1 (siTOPBP1 #1 and siTOPBP1 #2). Cells were arrested in pro-metaphase with Nocodazole, fixed 1 h after 1 Gy of IR and stained for CIP2A and TOPBP1. **f** Quantification of the experiment in **e**. Number of CIP2A foci per cell was assessed (siCtrl: $n = 62$, siTOPBP1 #1: $n = 46$, siTOPBP1 #2: $n = 44$, pooled from three independent experiments). Bars and error bars represent mean and SD. Statistical significance was assessed with the Kruskal-Wallis test and Dunn's multiple comparison test ($\alpha = 0.05$). All scale bars = 10 μm. Source data are provided as a Source Data file.

Fig. 6). Thus, our data suggest that CIP2A's role in the recruitment of TOPBP1 to sites of DSBs in mitosis is unlikely to be related to its PP2A inhibitory function.

**CRM1-dependent nuclear export sequesters CIP2A from TOPBP1 in interphase cells.** In a proteomic screen, CIP2A was identified as a target of the chromosome region maintenance 1 (CRM1, also called exportin 1) transport receptor for the export of proteins from the nucleus[27]. Indeed, we observed that treatment of cells with two selective CRM1 inhibitors (Leptomycin B and Selinexor) led to a significant increase in nuclear localization of CIP2A in interphase cells (Fig. 7a, b). CRM1 binds to its targets

via a flexible recognition motif called nuclear export signal (NES). Several NES were predicted in CIP2A based on the Eukaryotic Linear Motive Resource for Function Sites in Proteins (ELM[28]). One of them (amino acids 598–612) is localized between the ArmRD and the predicted C-terminal coiled coil region (Supplementary Fig. 7a), and deletion of a fragment comprising this sequence motif (amino acids 561–625; ΔNES) increased nuclear CIP2A concentration in interphase cells, indicating that CIP2A may be a direct CRM1 ligand (Fig. 7c, d Supplementary Fig. 7b). Indeed, CRM1 co-immunoprecipitated with overexpressed Flag-tagged CIP2A wild type and with Flag-CIP2A ΔNES, albeit with slightly reduced efficiency, suggesting the existence of one or

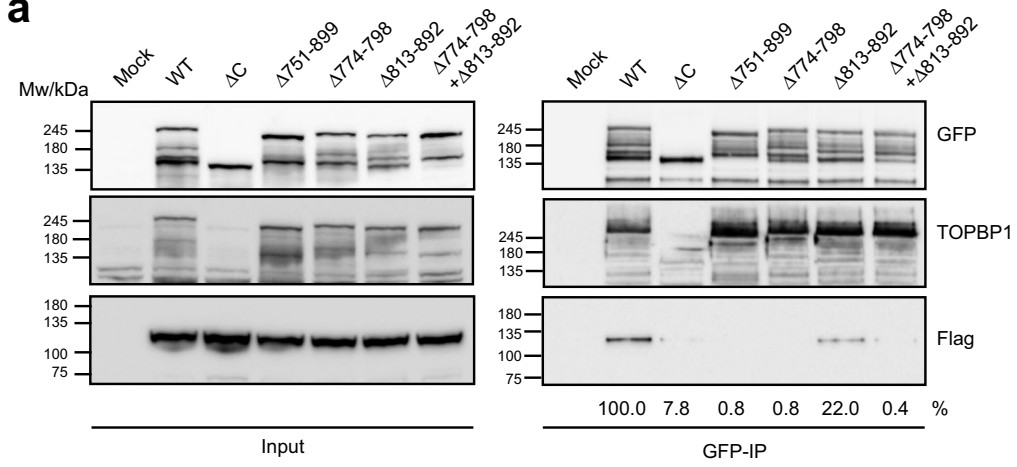

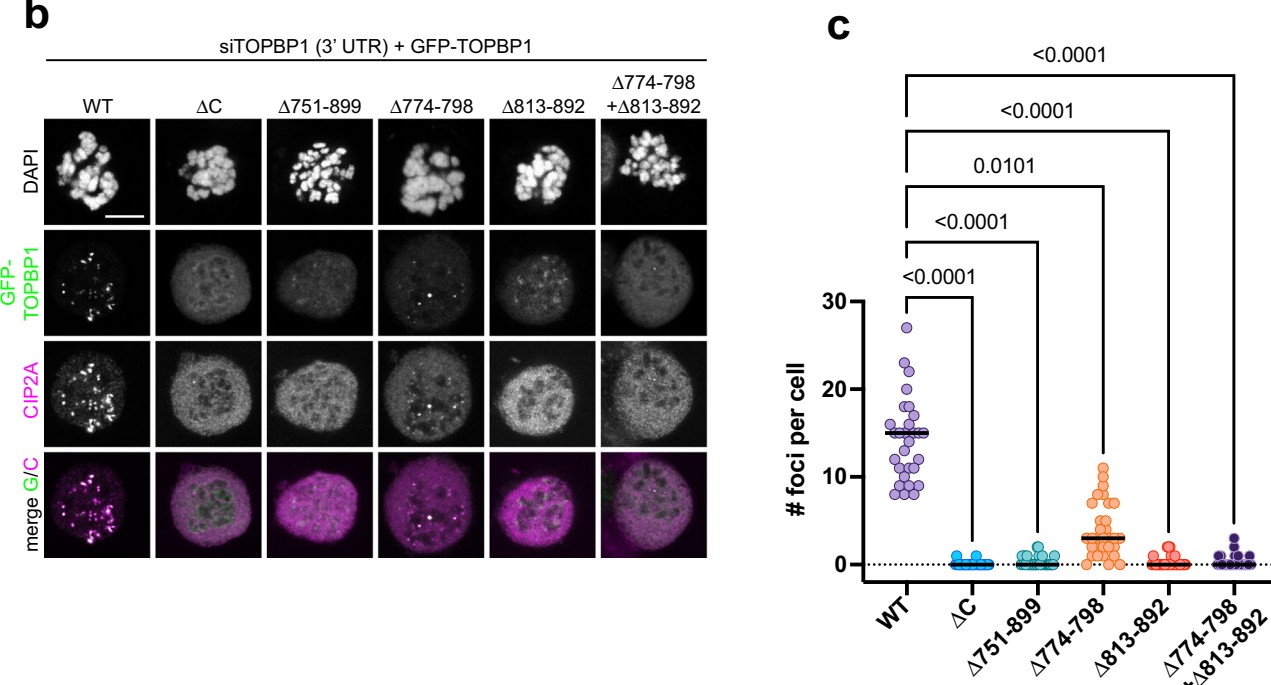

**Fig. 4 Two conserved sequence segments in TOPBP1 mediate its interaction with CIP2A. a** GFP-immunoprecipitation from 293FT cells either Mock transfected (Mock) or co-transfected with a GFP-tagged full-length TOPBP1 wild type (WT), and various deletion mutants as indicated, and Flag-tagged CIP2A. Relative intensities of co-immunoprecipitated Flag-CIP2A bands are indicated. **b** Confocal micrographs (maximum intensity projections) of Nocodazole-arrested GFP-TOPBP1 (WT and deletion mutants) expressing U2OS cells, treated with 1 Gy of IR. Endogenous TOPBP1 was depleted by 3'-UTR targeting TOPBP1 siRNA. **c** Quantification of GFP-TOPBP1/CIP2A foci per cell. Each data point represents one mitotic cell (WT: $n = 29$, ΔC: $n = 30$, Δ751–899: $n = 31$, Δ774–798: $n = 31$, Δ813–892: n = 30, Δ774–789 + Δ813-892: $n = 32$, pooled from three independent experiments). Statistical significance was assessed with the Kruskal-Wallis test and Dunn's multiple comparison test (α = 0.05). All scale bars = 10 μm. Source data are provided as a Source Data file.

several additional NES in CIP2A (Fig. 7e). Since TOPBP1 is predominantly nuclear, the active transportation of CIP2A from the nucleus to the cytoplasm suggests that these two proteins may be spatially separated during interphase. Staining of CIP2A in cells expressing endogenously tagged Lamin A (LMNA, a component of the nuclear envelope) showed that CIP2A is not present on chromatin until nuclear export ceases upon nuclear envelope breakdown at the onset of mitosis (Fig. 7f). Consistent with the idea that CIP2A is largely sequestered from TOPBP1 in interphase cells, we measured significantly increased PLA signals with CIP2A and TOPBP1 antibodies in mitotic cells (Fig. 7g and

Supplementary Fig. 7c). Together, these data suggest that CIP2A-TOPBP1 assembly is confined to mitosis through spatial separation of the two interaction partners in interphase cells by CRM1-mediated nuclear export of CIP2A.

**CIP2A is dispensable for TOPBP1 recruitment in interphase.** An important prediction from the spatial separation of CIP2A from TOPBP1 in interphase cells is that CIP2A is not implicated in TOPBP1 recruitment to sites of DSBs in interphase. Indeed, in response to DSB induction by IR, TOPBP1 foci formation in

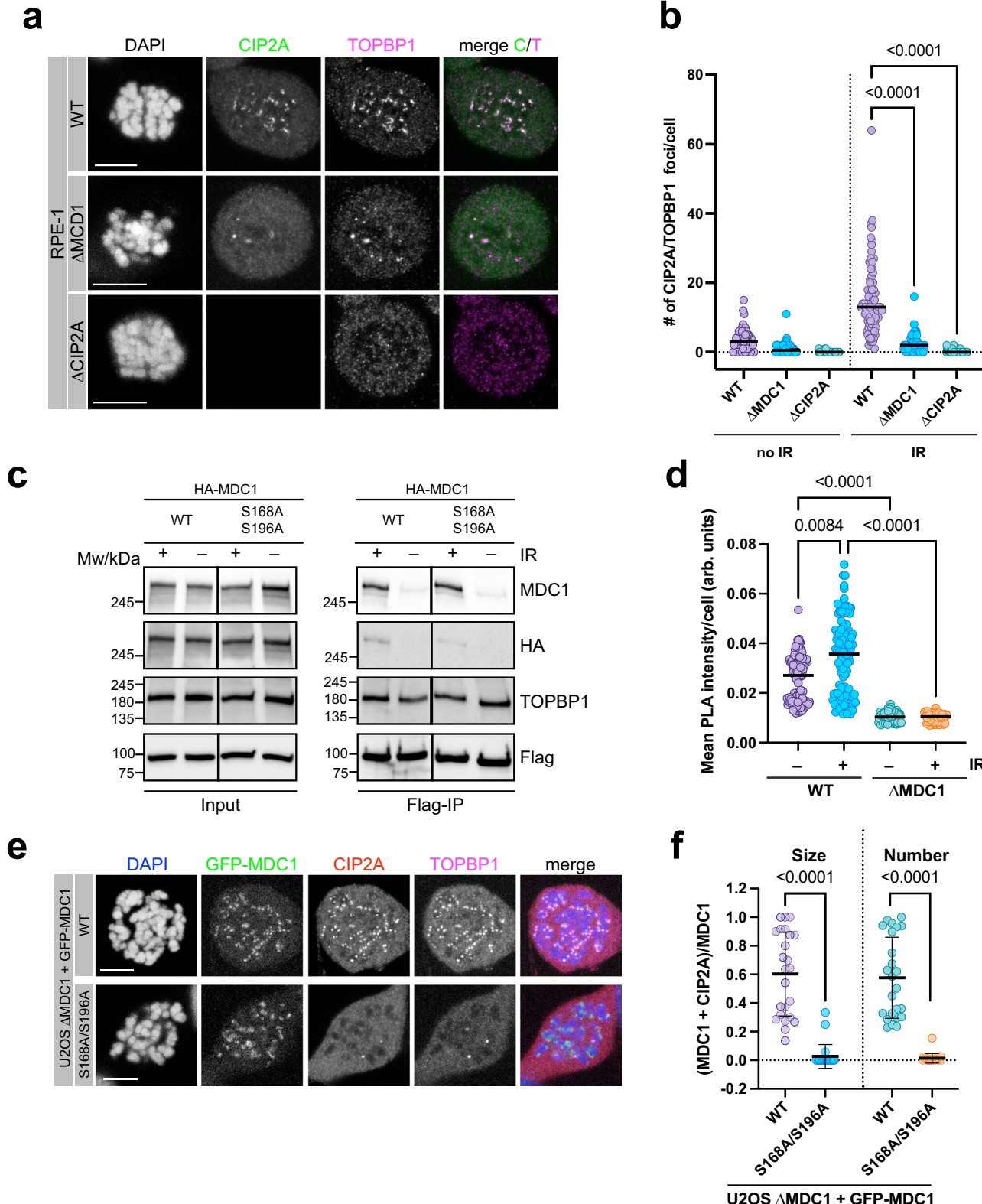

CIP2A deficient interphase RPE-1 cells is indistinguishable from cells that express CIP2A (Fig. 8a, b). Moreover, CIP2A is also not implicated in Treacle-mediated TOPBP1 accumulation in nucleolar caps after targeted induction of rDNA breakage (Supplementary Fig. 8a, b). Interestingly, we observed recruitment of CIP2A to foci and co-localization with TOPBP1 in a subset of Leptomycin B or Selinexor treated interphase cells (Supplementary Fig. 8c). This suggests that CIP2A, when forced in the nucleus in interphase cells, is capable to bind to TOPBP1 at sites of DSBs, which argues against a direct regulation of CIP2A-TOPBP1 association during the cell cycle. In support of this idea, TOPBP1 overexpression triggered nuclear condensate formation in interphase cells as previously reported[15,29,30]. These condensates contained significant amounts of CIP2A, but only when the CIP2A-interaction region (amino acid

**Fig. 5 CIP2A-TOPBP1 recruitment to sites of mitotic DSBs is mediated by MDC1. a** Confocal micrographs (maximum intensity projections) of Nocodazole-arrested RPE-1 wild type, MDC1 knock-out (ΔMDC1) and CIP2A knock-out (ΔCIP2A) cells, irradiated with 1 Gy and stained for TOPBP1 and CIP2A. **b** Quantification of the experiment in **a**. CIP2A/TOPBP1 foci were manually counted. Each data point represents one mitotic cell (no IR WT: $n = 52$, ΔMDC1: $n = 56$, ΔCIP2A: $n = 51$, IR WT: $n = 80$, ΔMDC1: $n = 61$, ΔCIP2A: $n = 53$, pooled from three independent experiments) and bars represent the median. Statistical significance was assessed with the Kruskal-Wallis test and Dunn's multiple comparison test ($α = 0.05$). **c** Flag-immunoprecipitation from 293FT cells transfected with Flag-tagged full-length CIP2A and HA-tagged MDC1 wild type (WT) and Ser168/Ser196 double mutant (S168A/S196A). **d** Quantification of CIP2A-MDC1 proximity by in situ PLA in U2OS wild type (WT) and U2OS MDC1 knock-out cells (ΔMDC1), arrested in mitosis by Nocodazole and mock treated or treated with 1 Gy of IR. Each data point represents one cell (WT -IR: $n = 131$, WT + IR: $n = 114$, ΔMDC1 -IR: $n = 85$, ΔMDC1 + IR $n = 54$, pooled from two independent experiments), and bars represent median. Statistical significance was assessed by Kruskal-Wallis test and Dunn's multiple comparison test ($α = 0.05$). **e** Confocal micrographs (maximum intensity projections) of Nocodazole-arrested U2OS ΔMDC1 cells stably transfected with GFP-tagged wild type and S168A/S196A mutated MDC1, stained for CIP2A and TOPBP1 1 h treatment with 1 Gy of IR. **f** Quantitative analysis of GFP-MDC1 and CIP2A co-localization by SQUASSH: Left graph: object size colocalization (area of object overlap divided by total object area). Right graph: object number colocalization (fraction of objects in each channel that overlap ≥ 50%). Data points represent individual mitotic cells ($n = 24$, except S168A/196 A object number: $n = 21$, pooled from three independent experiments). Bars and error bars represent mean and SD. Statistical significance was assessed by unpaired t-tests with Welch's correction ($α = 0.05$). Source data are provided as a Source Data file.

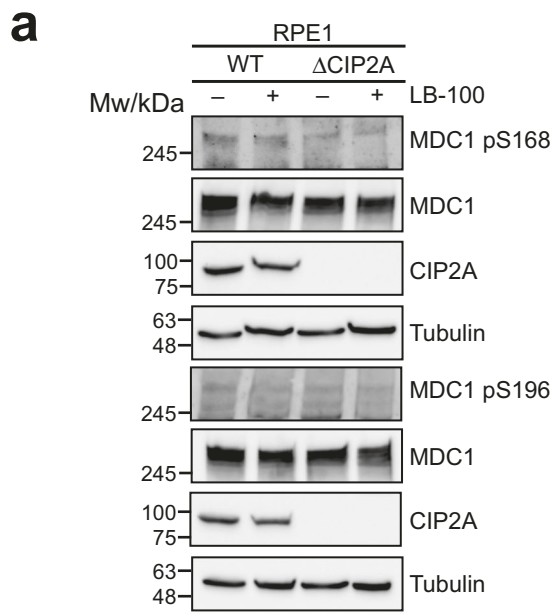

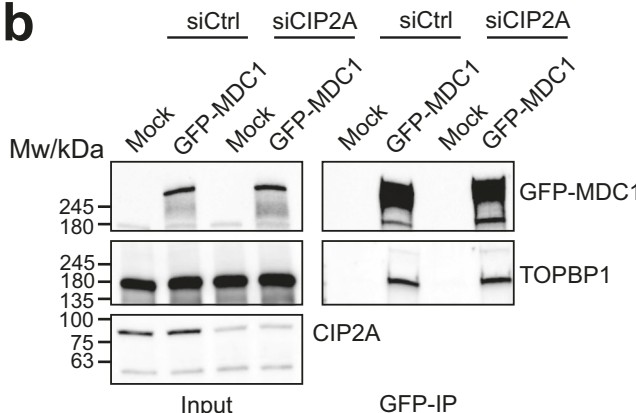

**Fig. 6 CIP2A controls TOPBP1 recruitment to sites of mitotic DSBs independently of PP2A. a** Western blots of total cell extract of RPE-1 wild type cells (WT) and RPE-1 CIP2A knock-out cells cells (ΔCIP2A) either Mock treated (-) or treated with the PP2A inhibitor LB-100. **b** GFP pull-down from 293FT cells transfected with GFP-tagged full-length MDC1 and either control siRNA (siCtrl) or siRNA against CIP2A (siCIP2A). All scale bars = 10 μm. Source data are provided as a Source Data file.

751–899) was present in the overexpressed TOPBP1 (Supplementary Fig. 8d). These data show that CIP2A regulates TOPBP1 exclusively in mitotic cells while it is dispensable for TOPBP1 recruitment in interphase. Moreover, the data suggest that CIP2A can constitutively interact with TOPBP1 throughout the cell cycle (and in post-lysis cell extracts) and that the interaction between these two proteins is thus primarily regulated by their spatial segregation in interphase cells.

**Loss of CIP2A impairs maintenance of chromosomal stability during mitosis.** Cells unable to recruit TOPBP1 in mitosis due to disrupted MDC1-TOPBP1 interaction show increased radio-sensitivity, micronuclei (MNi) formation and chromosomal instability[11], most likely due to the inability to properly segregate acentric chromosome fragments[31]. In line with its capacity to control TOPBP1 localization in mitosis, CIP2A deficient cells also displayed increased radio-sensitivity and spontaneous MNi formation, which was slightly increased after IR treatment and was rescued by re-expression of wild type CIP2A (Fig. 9a, b). The majority of MNi in CIP2A deficient cells stained negative for the centromeric marker CENPA and positive for the DSB marker γH2AX, suggesting that they mostly contain acentric chromosome fragments (Fig. 9c and Supplementary Fig. 9a, b). To explore whether loss of CIP2A affects DSB repair in the subsequent interphase after irradiation in mitosis, we first synchronized parental RPE-1 cells and ΔCIP2A cells as well as ΔCIP2A cells stably expressing Flag-tagged wild type CIP2A in mitosis followed by irradiation and release from the mitotic block. γH2AX foci were quantified 24 h post-irradiation. There was a significant increase in residual γH2AX foci in cells deficient for CIP2A 24 h post-irradiation and re-expression of wild type CIP2A rescued this phenotype (Fig. 9d and Supplementary Fig. 9c). We also observed a significantly increased number of abnormal structures in metaphase spreads of RPE-1 ΔCIP2A cells, but not in RPE-1 ΔCIP2A cells stably transduced with wild type CIP2A (Fig. 9e, f). We conclude from these results that in the absence of CIP2A, cells are unable to properly stabilize and/or repair DSBs during mitosis, which leads to an accumulation of MNi that contain acentric chromosome fragments, ultimately resulting in chromosomal instability.

**Discussion**
In this study we identified and characterized a highly conserved bi-partite protein interaction surface in TOPBP1 located between BRCT domains 5 and 6. An unbiased search for factors that interact with this region uncovered CIP2A as a new TOPBP1 interaction partner. We show that CIP2A and TOPBP1 are dependent upon each other for recruitment to sites of DSBs

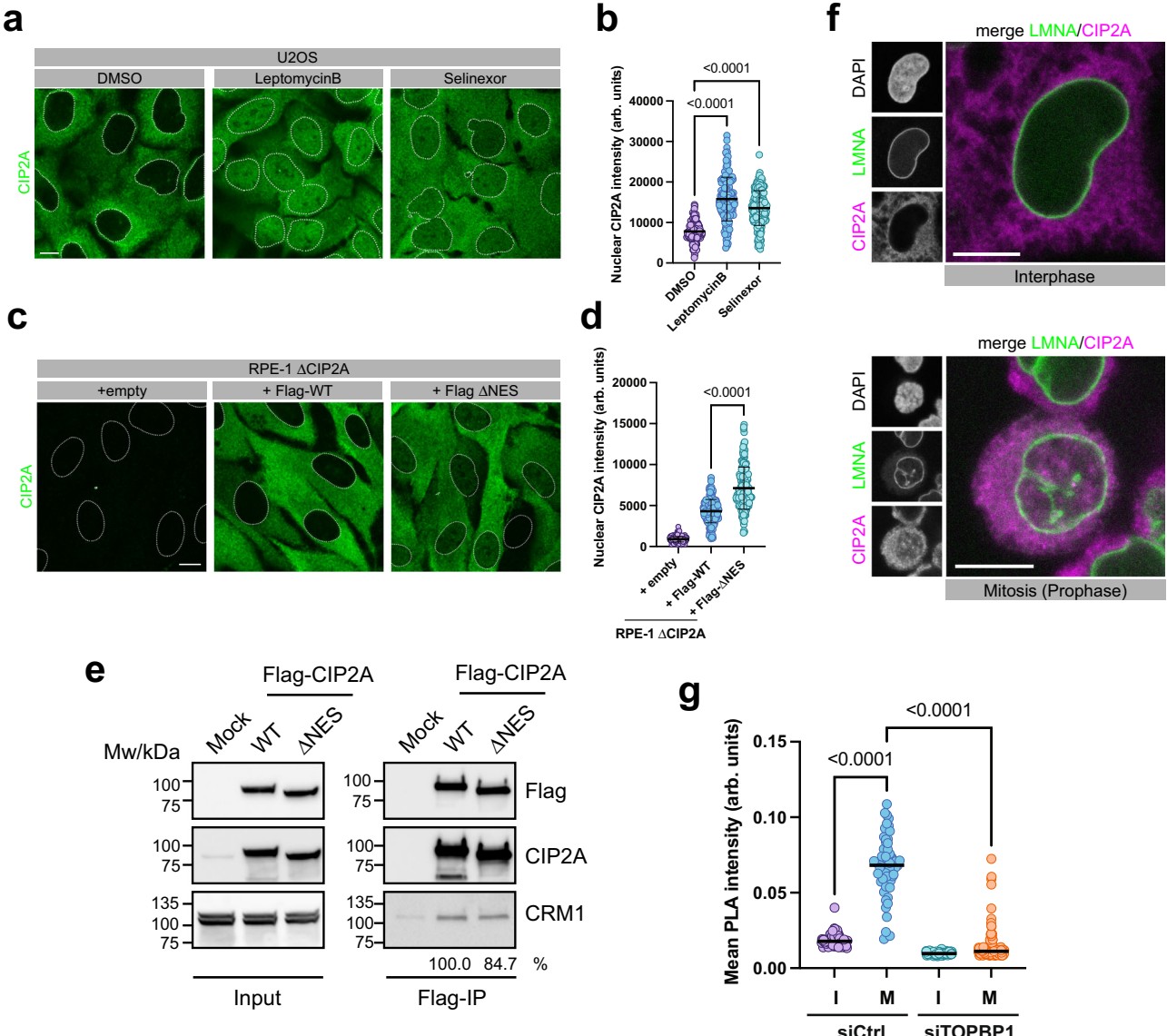

**Fig. 7 CRM1-dependent nuclear export sequesters CIP2A from TOPBP1 in interphase cells. a** Confocal micrographs (maximum intensity projections) of Leptomycin B, Selinexor and control DMSO treated U2OS cells, stained for CIP2A. **b** Quantification of nuclear CIP2A staining of the experiment in **a**. Each data point corresponds to one cell (DMSO: $n = 173$, LeptomycinB: $n = 150$, Selinexor: $n = 178$, pooled from two independent experiments). Bars and error bars represent mean and SD. Statistical significance was calculated using one-way ANOVA and Dunnett's multiple comparison test ($\alpha = 0.05$). **c** Confocal micrographs (maximum intensity projections) of RPE-1 $\Delta$CIP2A cells and RPE-1 $\Delta$CIP2A cells stably transduced with Flag-tagged full-length CIP2A (WT) and CIP2A deletion mutant (human CIP2A amino acids 561–625; $\Delta$NES), stained for CIP2A. **d** Quantification of nuclear CIP2A staining of the experiment in **c**. Each data point corresponds to one cell (+Epmpty: $n = 199$, +Flag-WT: $n = 155$, +Flag-$\Delta$NES: $n = 165$, pooled from two independent experiments). Bars and error bars represent mean and SD. Statistical significance between +Flag-WT and +Flag-$\Delta$NES was calculated using two-sided Welch's $t$-test ($\alpha = 0.05$). **e** Flag-immunoprecipitation from 293FT cells either Mock transfected (Mock) or transfected with a Flag-tagged full-length CIP2A wild type (WT), and Flag-tagged CIP2A lacking amino acids 561-625 ($\Delta$NES). Relative intensities of co-immunoprecipitated CRM1 bands are indicated. **f** Confocal micrograph (maximum intensity projection) of Hela cells expressing endogenous Clover-tagged LMNA and stained for CIP2A. **g** Quantification of CIP2A-TOPBP1 proximity by in situ PLA in U2OS cells transfected with control siRNA (siCtrl) and siRNA against TOPBP1 (siTOPBP1), arrested in mitosis by Nocodazole and treated with 1 Gy of IR. Interphase cells (I) were separated from mitotic cells (M) with an automatic image analysis pipeline, based on DAPI mean intensities. Each data point represents one cell (siCtrl I: $n = 92$, siCtrl M: $n = 63$, siTOPBP1 I: $n = 138$, siTOPBP1 M: $n = 121$; pooled from two independent experiments), and bars represent median. Statistical significance was assessed by Kruskal-Wallis test and Dunn's multiple comparison test ($\alpha = 0.05$). Source data are provided as a Source Data file.

specifically during mitosis. Deletion of two highly conserved amino acid patches within the CIP2A-interacting region in TOPBP1 abrogated interaction with CIP2A and led to a complete DSB recruitment defect in mitotic cells. These findings suggest that CIP2A and TOPBP1 must exist in a complex for their efficient recruitment to sites of mitotic DNA breaks. Surprisingly,

deletion of the longer, more C-terminal conserved patch alone (aa 813-892) also completely abrogated recruitment of TOPBP1 to sites of damage, while its interaction with CIP2A was only partially defective. This is somewhat in conflict with two recently published studies in which yeast-two-hybrid analysis was used to map the CIP2A interaction site in TOPBP1 to amino acids

**a**

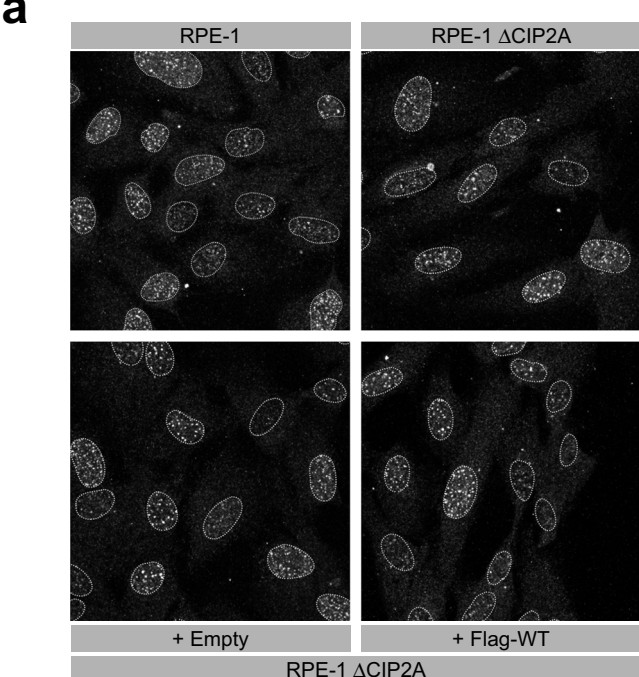

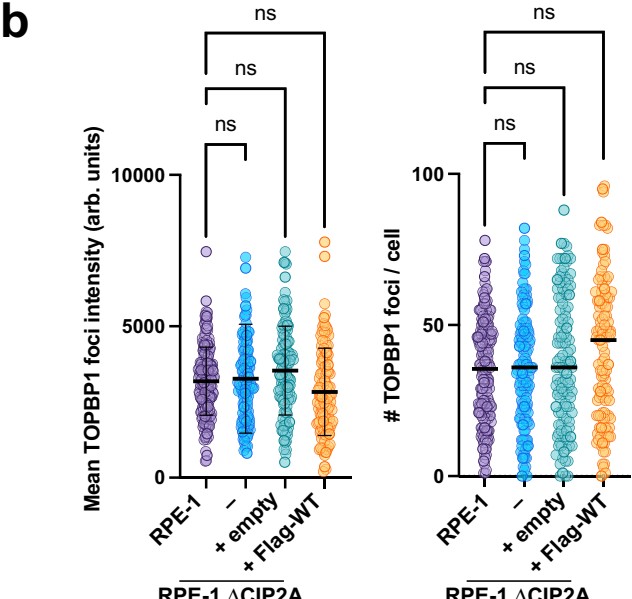

**b**

### Fig. 8 CRM1 is dispensible for TOPBP1 recruitment in interphase.

**a** Confocal micrographs of unsynchronized of parental RPE-1 cells (RPE-1), RPE-1 CIP2A knock-out cells (RPE-1 ΔCIP2A) and RPE-1 ΔCIP2A cells stably transduced with empty vector (+ Empty), Flag-tagged wild type CIP2A (+ Flag-WT), treated with 3 Gy of IR and stained for TOPBP1. **b** Quantification of TOPBP1 foci mean intensity and number of TOPBP1 foci per nucleus in interphase parental RPE-1 cells, RPE-1 CIP2A knock-out cells (ΔCIP2A) and ΔCIP2A cells stably transduced with empty vector (+Empty) or Flag-tagged full-length CIP2A (+ Flag-WT). Each data point represents one cell (RPE-1: $n = 200$; ΔCIP2A: $n = 144$; +Empty: $n = 128$; +Flag-WT: $n = 125$; pooled from two independent experiments) bars and error bars left graph represent mean and SD, bars right graph represent median. Statistical significance in the left graph (mean intensities) was assessed by one-way ANOVA and Dunnett's multiple comparison test. Statistical significance in the right graph (foci number) was assessed by Kruskal-Wallis test and Dunn's multiple comparison test ($\alpha = 0.05$) All scale bars = 10 µm. Source data are provided as a Source Data file.

is not stimulated by DNA damage (see Figs. 1e and 2f). However, interaction between CIP2A and MDC1 is stimulated by DNA damage, both in vitro and in cells (see Fig. 5c, d). Thus, TOPBP1, despite being able to interact with both MDC1 and CIP2A in a constitutive manner (this study and ref. [10]), does not appear to simply bridge the two proteins to form a stable ternary complex. Instead, the formation of a ternary complex occurs via a process that is stimulated by DNA damage. How this works on a mechanistic level is currently not clear, but one possibility is that in the presence of DNA damage, CIP2A and TOPBP1 may form large multimeric assemblies. Under physiological conditions, CIP2A and TOPBP1 accumulation is strictly dependent on DNA damage. However, we and others recently demonstrated that when TOPBP1 is overexpressed and/or forced to self-associate, accumulation can occur even in the absence of DNA damage[15,29,30]. It is therefore possible that DNA damage may stimulate CIP2A and TOPBP1 multimerization, which may be a prerequisite for their efficient association with MDC1 at sites of mitotic DSBs.

In summary, our findings demonstrate that CIP2A functions downstream of MDC1 to mediate TOPBP1 recruitment to sites of chromosome breaks in mitosis, which is critical for proper segregation of acentric chromosome fragments. As CIP2A deficiency gives rise to DSB repair defects after irradiation of mitotic cells, it is possible that CIP2A mediates productive DSB repair by stabilizing and/or tethering broken chromosomes during cell division until they enter the subsequent cell cycle where these lesions can be repaired.

The effect of CIP2A on TOPBP1 is restricted to mitosis by nuclear exclusion, which is similar to the spatial control of the GEN1 Holliday junction resolvase whose activity is needed at the onset of mitosis but causes elevated crossover formation when present in the nucleus in interphase[32]. Cancer cells must frequently deal with remnants of replication problems from the preceding S-phase, either because of oncogene-driven cell cycles, or as a result of defective HR pathways due to the loss of tumor suppressor genes such as BRCA1 and BRCA2. We surmise that at least a subset of these replication intermediates will be converted to DSBs in mitosis and will thus require the CIP2A-TOPBP1 complex for their stabilization. Indeed, in a complementary work, we found that loss of CIP2A or disruption of the CIP2A-TOPBP1 complex is lethal in HR-deficient cells due to rampant mis-segregation of acentric chromatin fragments that occur because of aberrant or incomplete DNA replication in these cells[18]. Collectively, our data firmly establish CIP2A-TOPBP1 as a physiologically critical component of the cellular machinery that deals with DNA lesions specifically during mitosis.

830–852[17,18]. An explanation for this discrepancy may be the different assays used. It is thus possible that the residual binding activity upon deletion of aa 813–892 observed here may be caused by protein over-expression in the co-immunoprecipitation assay. On the other hand, deletion of the more N-terminal, shorter conserved patch (aa 774–789) also resulted in defective interaction with CIP2A and partially abrogated DSB recruitment, thus indicating that sequences beyond the minimal interaction region mapped by yeast-two-hybrid analysis also contribute to CIP2A-TOPBP1 association and efficient DSB recruitment. Further biochemical and structural analysis will be required to reveal the details of the interaction mechanism.

We provide strong evidence that CIP2A and TOPBP1 need to form a ternary complex with MDC1 to be efficiently recruited to sites of DSBs in mitosis. Surprisingly, both in vitro analysis and PLA assays revealed that interaction between TOPBP1 and CIP2A

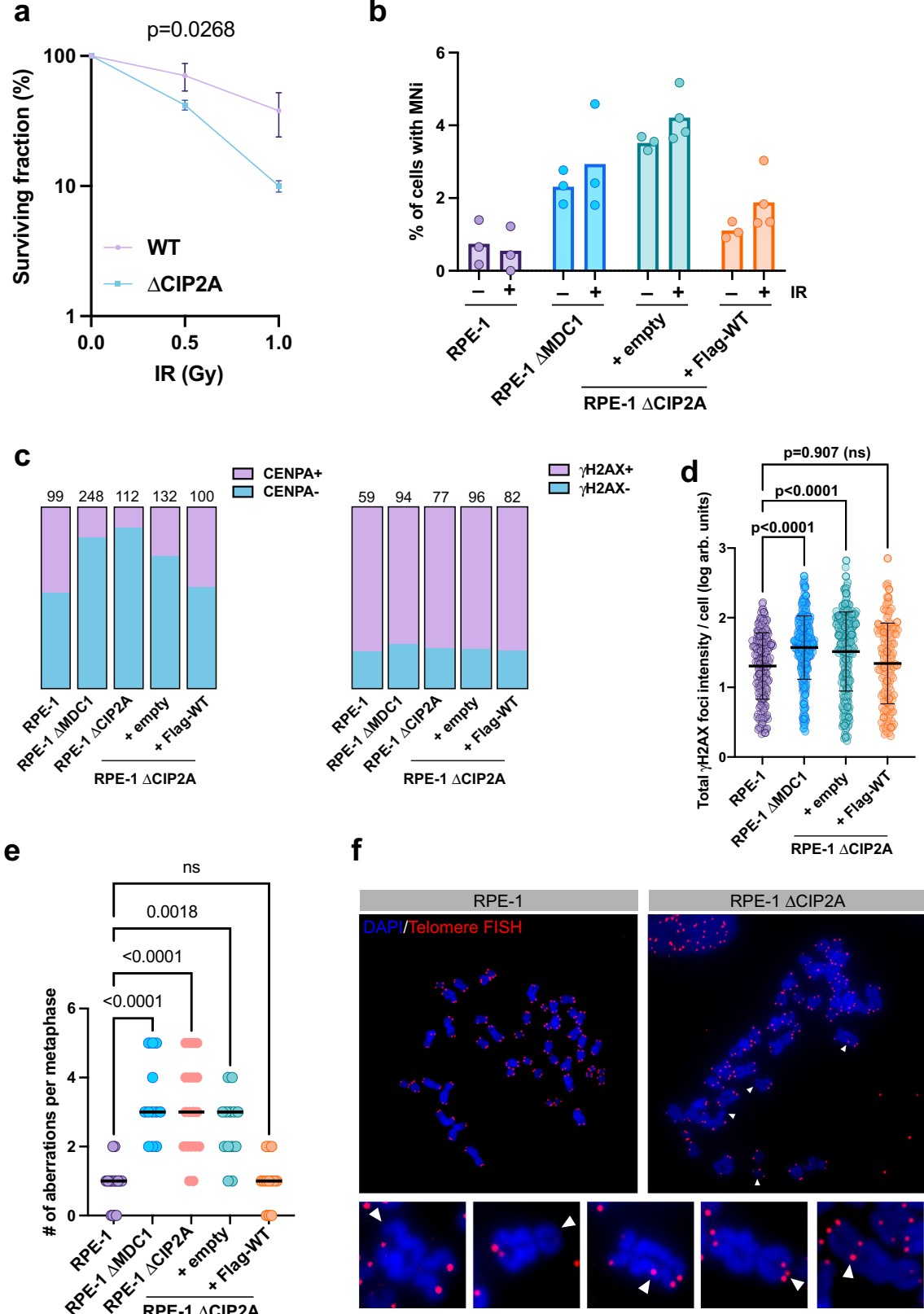

# Methods

**Cell culture and drug treatments**. All cell lines were grown in a sterile cell culture environment and were routinely tested for mycoplasma contamination. 293FT (transformed human embryonal kidney cell line), U2OS (human osteosarcoma cell line), RPE-1 (hTERT immortalized human retinal pigmented cell line) and HeLa (human cervical carcinoma cell line) cells were cultured in Dulbecco's modified Eagle's medium (DMEM Gibco™, 61965026), DLD1 (human colon carcinoma cell line) cells were cultured in RPMI-1640 medium (ATCC 30–2001). All cell culture medium was supplemented with 10% fetal bovine serum (FBS), 2 mM L-glutamine and penicillin-streptomycin antibiotics. Cell lines were grown under standard cell culture conditions in $CO_2$ incubators (37 °C; 5% $CO_2$). Genetically modified U2OS cell lines U2OS ΔMDC1, U2OS ΔMDC1 + GFP-MDC1 WT and U2OS ΔMDC1 + GFP-MDC1 S168A/S196A were described[11]. HeLa and U2OS cells in which the nuclear lamina protein LMNA was endogenously tagged with the

**Fig. 9 Loss of CIP2A impairs maintenance of chromosomal stability during mitosis. a** Clonogenic survival analysis of IR treated RPE-1 wild type (WT) cells and RPE-1 CIP2A knock-out cells (ΔCIP2A). Data points represent the mean of 3 independent experiments, error bars represent the SD. Statistical significance was assessed by linear regression (α = 0.05). **b** Quantification of MNi formation, **c** quantification of CENPA positive (CENPA + ) and CENPA negative (CENPA-) as well as γH2AX positive (γH2AX + ) and γH2AX negative (γH2AX-) MNi, **d** quantification of residual γH2AX foci 24 h after irradiation of mitotic cells and **e** quantification of chromosomal aberrations in metaphase spreads in RPE-1 parental cells (RPE-1), RPE-1 MDC1 knock-out cells (RPE-1 ΔMDC1), RPE-1 CIP2A knock-out cells (RPE-1 ΔCIP2A) and RPE-1 CIP2A knock-out cells stably transduced with empty vector (+ empty) and Flag-tagged wild type CIP2A (+ Flag-WT). Columns in **b** represent mean of 3–4 independent experiments (RPE-1 ± IR, RPE-1 ΔMDC1 ± IR, + empty -IR, + Flag-WT -IR: n = 3, +empty +IR, + Flag-WT + IR: n = 4), data points represent percentage of cells with MNi in the experiments. Parts of whole are displayed for each cell line in **c**. The numbers of MNi assessed for each cell line (n) in **c** are indicated above each column. Each data point in **d** represents log of total γH2AX foci intensity per cell (RPE-1: n = 368; RPE-1 ΔMDC1: n = 330; +Empty: n = 219; +Flag-WT: n = 224), bars and error bars represent mean and SD. Statistical significance was assessed by one-way ANOVA and Sidak's multiple comparison test (α = 0.05) ns: not significant. Aberrations in **e** were counted manually. Each data point represents one metaphase (RPE-1: n = 16, RPE-1 ΔMDC1: n = 12, RPE-1 ΔCIP2A: n = 19, +Empty: n = 13, +Flag-WT: n = 13), bars represent the median. Statistical significance was assessed by the Kruskal-Wallis test and Dunn's multiple comparison test (α = 0.05). ns: not significant. **f** Examples of chromosomal aberrations in metaphase spreads derived from RPE-1 parental (RPE-1) and RPE-1 CIP2A knock-out cells (ΔCIP2A). Aberrations including single chromatid telomere loss, sister chromatid telomere loss, interstitial telomeres, telomere duplications and dicentric chromosomes are indicated with white arrowheads. For **d**, **e**: One representative of two independent experiments is shown. Source data are provided as a Source Data file.

monomeric green fluorescent protein Clover were generated as described[33]. DSB reporter cell line U2OS ER-mCherry-LacI-FokI-DD was described[26]. RPE-1 ΔMDC1 and RPE-1 H2AX[S139A/S139A] were described[34]. Stably transfected U2OS cell lines were cultured in the presence of 400 μg/mL Geneticin (G418). For cell synchronization in pro-metaphase, cells were incubated with 5 μg/ml Nocodazole (Sigma-Aldrich, M1404) for 16 h. For CRM1 inhibition, 10 μM Leptomycin B (Apollo Scientific) or 10 μM Selinexor (Focus Biomolecules) were added to the medium for 3 h. For PP2A inhibition, 10 nM LB-100 (Selleckchem, S7537) was added for 2 h. mCherry-Fok1 DSB-reporter was induced by treatment with 1 μM Shield-1 and 1 μM 4-hydroxytamoxifen (4-OHT) for 4 h. Irradiation of cells was done in a YXLON, Y.SMART583 custom-made X-ray device or a Faxitron 43855D irradiator.

**Cloning and mutagenesis**. pIRESneo2-EGFP-TOPBP1 plasmids are described in[13]. pIRESneo2-EGFP-TOPBP1-ΔBRCT6 and pIRESneo2-EGFP-TOPBP1-Δ774-798 were generated by site directed mutagenesis of pIRESneo2-EGFP-TOPBP1-WT using QuikChange II XL Site-Directed Mutagenesis kit (Agilent Technologies, 200522). Sequences of the mutagenesis primers for TOPBP1-ΔBRCT6 were, forward: 5'-CCC AAA ATG AGC TTG GAT ATC AGC GCA-3'; reverse:

5'- TTC CTT CTC TGA CTG GGC CTC TTT CAG-3', and for TOPBP1-Δ774-798 were, forward: 5'-GCC TGC AAA CTC ACG CCA GAC AGG TCG C-3'; reverse: 5'-GCG ACC TGT CTG GCG TGA GTT TGC AGG C-3'. Generation of pIRESneo2-EGFP-TOPBP1-W1145R was described[15]. pIRESneo2-EGFP-TOPBP1-Δ751-899, pIRESneo2-EGFP-TOPBP1-Δ813–892 and pIRESneo2-EGFP-TOPBP1-Δ774-798 + Δ813-892 were generated by cloning synthetic fragments (synthesized by Genscript) of TOPBP1, lacking the codons of the indicated amino acids, into pIRESneo2-EGFP-TOPBP1-WT using SbfI-HF® (NEB, R3642S) and StuI (NEB, R0187S) restriction enzymes. pcDNA3.1(+)-N-HA-TOPBP1(740-899) was custom-synthesized by Genscript. pHIV-NAT-T2A plasmid containing Flag-tagged full-length CIP2A cDNA is described[18]. Deletion of the residues 561-625 (ΔNES) was done by deletion PCR using Phusion® High-Fidelity DNA Polymerase (NEB), followed by DpnI digestion of the template and transformation into bacteria. Primers were as follows: forward: 5'-AAA CAA TGC CTA TAG ACA ACA GGA ATA TGA AAT GAA ACT ATC CAC ATT AG-3'; reverse:

5'- CTA ATG TGG ATA GTT TCA TTT CAT ATT CCT GTT GTC TAT AGG CAT TGT TT-3'. Plasmids pIRES V5 I-Ppo1 for I-Ppo1 mRNA production and purification were described[35]. For CIP2A ArmRD expression in bacteria, bases one through 1677 (559 aa) of the full length CIP2A cDNA template were amplified by PCR and were subsequently cloned into the bacterial expression vector pGEX-4T3 by In-Fusion® cloning (Takara). PCR primers were: forward: 5'-GTG GAT CCC CGA ATT GGA CTC CAC TGC CTG CTG AG-3', reverse: 5'- GTC GAC CCG GGA ATT TCT GTT GCT ATA GGC ATT GTT GC-3'.

**Generation of a CIP2A knock-out RPE-1 cell line by CRISPR/Cas9**. RPE-1 cells were cultured overnight at a density of 0.1 × 10^6. For transfection, 375 ng of synthetic sgRNA (TrueGuide™, CRISPR932782_SGM; sgRNA sequence: GGTCTTCAAGTAGCTCTACA) along with 6 μL Cas9 mRNA (Invitrogen), and 18 μL of the MessengerMAX™ reagent (Invitrogen) were separately diluted in 300 μL Opti-MEM™ I (Gibco) each. After 10 min of incubation at room temperature, both mixtures were combined and added to the plate at 100 μL per well. The cells were kept in culture for another 48 h, trypsinized and transferred to a 96-well plate in serial two-fold dilution, starting with 4,000 cells. Wells containing single colonies were identified under a transluminescence microscope and progressively expanded to larger culture volumes. One successful knock-out clone could be identified by means of immunofluorescence as well as western blotting of total cell extracts. A bi-allelic single dA insertion at position 5110 of the gene was

confirmed by subcloning the targeted area into a pCR™4Blunt-TOPO® vector (Invitrogen) and subsequent colony sequencing of transformed Stellar™ Competent E. coli cells (Takara).

**Complementation of RPE-1 ΔCIP2A cells by lentiviral transduction**. For lentivirus production 293FT cells were plated at 80% confluence in a 10 cm dish with 8 mL of culture medium one day prior to transfection. Transfection was carried out using Lenti-X Packaging Single Shot (VSV-G) system (Takara Bio, 631275) according to the manufacturer's protocol. 7 μg of pHIV-NAT-T2A empty plasmid or plasmid containing Flag-tagged CIP2A cDNA (WT and ΔNES) were used. Lentivirus-containing supernatants were centrifuged at 4 °C for 10 min at 500 g and passed through a 0.45 μm filter. Virus titer was measured using the QuickTiter™ Lentivirus Titer Kit (Cell Biolabs, VPK-107-T) according to the manufacturer's instructions. For lentiviral transduction, RPE-1 ΔCIP2A cells were plated in 6 cm dishes at 80% confluence in culture medium one day prior to infection. Cells were incubated with virus-containing supernatants in the presence of 8 μg/mL Polybrene (Santa Cruz Biotech) for 24 h. The virus volume needed for infection was calculated using the following formula: Total Transduction Units (TU) needed = cells seeded*MOI (MOI = 5 was used). After incubation, virus-containing medium was discarded and replaced with fresh culture medium. Cells were incubated for an additional 48 h before proceeding with protein analysis to evaluate expression of the gene of interest. Transduced cells were selected in the presence of 100 μg/mL Nourseothricin dihydrogen sulfate (Roth, 3011). Stably transduced cell lines were cultured in the presence of 50 μg/mL Nourseothricin dihydrogen sulfate.

**Cell transfection with RNA and DNA**. The control siRNA (siCtrl), siRNA targeting TOPBP1 (siTOPBP1 #1) and the 3' UTR sequence of TOPBP1 gene (siTOPBP1 #2) were obtained from Microsynth AG. The sequences of the siRNAs are the following:
siCtrl: UGGUUUACAUGUCGACUAA-dTdT,
siTOPBP1 #1: ACAAAUACAUGGCUGGUUA-dTdT
siTOPBP1 #2: GUAAAUAUCUGAAGCGUUAUU-dTdT
ON-Target plus SMART pool siRNA targeting CIP2A was purchased from Dharmacon (L-014135-01-0005) and contained a mix of four different siRNAs:
Sequence 1: ACAGAACUCACACGACUA
Sequence 2: GUCUAGGAUUAUUGGCAAA
Sequence 3: GAACAAAGGUUGCAGAUUC
Sequence 4: GCAGAGUGAUAUUGAGCAU
For siRNA transfection, cells were grown in a six-well plate 24 h prior to transfection. A total of 60% confluent cells were transfected using 20 nM siRNA and Lipofectamine RNAiMAX™ (Invitrogen) according to the manufacturer's instructions. Cells were incubated with siRNA-lipid complexes for 48 h before being seeded on coverslips and harvested 72 h after transfection. I-Ppo1 mRNA transfection was done using the Lipofectamine MessengerMax™ Reagent (Invitrogen) according to the manufacturer's protocol.
Plasmid DNA was transfected using jetOPTIMUS® DNA Transfection Regent in 6-well plates following the standard protocol provided by the manufacturer. Cells were then incubated for 24 h before proceeding with further analysis.

**SDS-PAGE and Western blotting**. To prepare cell extracts for SDS-PAGE, cells were washed in PBS, scraped off the plate and pelleted before resuspension in RIPA lysis buffer (150 mM NaCl, 50 mM Tris-HCl pH 8, 1% Igepal CA-630, 0.5% sodium deoxycholate, 0.1% SDS). Lysis buffer was supplemented with cOmplete EDTA-free protease Inhibitor cocktail (Roche) before use. Next, cells were incubated on ice for 30 min and vortexed every 10 min to dissolve the cell clumps. After

incubation, lysates were centrifuged at 16,000 g for 15 min in a refrigerated centrifuge and clear lysates was transferred into a fresh tube for protein quantification. To detect phosphorylated proteins, SDS-lysis buffer (120 mM Tris pH 6.8, 20% glycerol, 4% SDS) was used to lyse cells directly on the plate. Protein aliquots were mixed with 2x SDS sample buffer (Geneaid, PLD001, 100 mM DTT) and heated for 5′ at 95 °C before loading on the gel.

SDS-PAGE was performed using 4–20% Mini-PROTEAN TGX Stain-free Precast Gels (Bio-Rad, 4568094) except for the experiments in Fig. 4a and Supplementary Fig. 4b, where mPAGE™ 4–20% Bis-Tris Precast Gels (Millipore, MP42G10) were used. As a reference for molecular weights, Prestained Protein Ladder 245 kDa (Geneaid) was loaded on the gel. Western blotting was performed using the Trans-Blot® Turbo™ Transfer system (Bio-Rad) on 0.2 μm nitrocellulose membranes. Membranes were blocked for at least 2 h with 5% milk in TBS-Tween. To detect phosphorylated proteins, membranes were blocked for 1 h with EveryBlot Blocking Buffer (Bio-Rad). Antibodies against the following proteins were used at the indicated dilutions: GFP (Mouse, Roche, 11814460001, 1:5000), GFP (Rabbit, Abcam, ab290, 1:1000), HA (Rabbit, Abcam, ab9110, 1:4000), MDC1 (Rabbit, Abcam, ab11171, 1:5000), MDC1 pSer168 (Rabbit, AMS Biotechnology, custom-made, 1:500)[11], MDC1 pSer196 (Rabbit, 21st Century, custom-made 1:200)[11], TOPBP1 (Rabbit, Abcam, ab2402, 1:1500), CIP2A (Rabbit, Cell Signaling, 14805, 1:1000), CIP2A (Mouse, Santa Cruz, sc-80659, 1:1000), CRM1 (Rabbit, Atlas Antibodies, HPA042933, 1:1500), Tubulin (Mouse, Sigma, T6199, 1:5000), Tubulin hFAB™ rhodamine (Bio-Rad, 12004166, 1:5000), FLAG M2 (Mouse, Sigma, F1804, 1:1000), pSer10 Histone H3 (Rabbit, Millipore, 06-570, 1:1000), Histone H3 (Rabbit, Abcam, ab1791, 1:10000). Blots were incubated with primary antibodies overnight at 4 °C and washed 3 × 10 min with TBS-Tween before incubation for 1 h at with secondary antibodies at room temperature: HRP-conjugated anti-mouse (GE Healthcare, NA934, 1:10000) or HRP-confugated anti-rabbit (GE Healthcare, NA934, 1:10000). Blots were developed with ECL™ Prime Western Blotting Detection Reagent (Cytiva), and image acquisition was done on a ChemiDoc MP Imaging System (Bio-Rad).

**Expression and purification recombinant proteins**. For purification of the CIP2A ArmRD, pGEX-4T3 containing the coding sequence of aa 1–559 of CIP2A was transformed into BL21(DE3)pLysS competent cells for high-efficiency expression of the recombinant protein under the IPTG inducible control of a T7 promoter. The harvested bacteria were lysed through repeated freezing (down to −80 °C) and thawing, followed by vigorous sonication in an ice-cooled buffer with 1% Triton X-100. Finally, the extract was cleared from insoluble debris, centrifuged for 30 min at 15,000 rcf and 4 °C, and then loaded onto a high performance 1 mL GSTrap™ affinity column (Cytiva, 17-5281-01) for purification on an NGC Quest 10 chromatography system (Bio-Rad). For expression and purification of the HA-tagged TOPBP1 aa740–899 fragment, ExpiCHO-S cells (Life Technologies) were cultured in suspension at 36 °C and 8% CO₂ using the CHOgro® Expression System (Mirus, MIR 6270) inside of 50 mL TubeSpin® bioreactors (TPP, 87050) while shaking at 225 rpm on a Rotamax 120 orbital shaker. The cells were adjusted to a density of $4 \times 10^6$/mL before transfection with a mix of the TransIT-PRO® reagent and pcDNA3.1(+)-N-HA-TOPBP1(740–899) plasmid DNA. After adding the complex, the cell suspension was further supplemented with CHOgro® titer enhancer and its temperature was lowered to 32 °C. A sample was collected for co-immunoprecipitation after seven days.

**Immunoprecipitations**. To prepare cell extracts for immunoprecipitation, cells were washed in cold PBS buffer pH 7.45, harvested with a cell scraper and centrifuged at 500 g for 3 min. PBS was then discarded and cells were resuspended in an appropriate volume of IP lysis buffer (50 mM Tris-HCl pH 7.6, 150 mM NaCl, 5 mM MgCl₂, 0.5% Triton X-100), supplemented with cOmplete EDTA-free protease inhibitor cocktail and 25 U/ml Benzonase nuclease (Sigma, E1014). Cells were incubated on ice for 30 min and lysates were cleared by centrifugation at 16,000 g (4 °C) for 15 min. For HA-immunoprecipitations, 30 μL of monoclonal anti-HA agarose beads (Sigma, A2095) were added to 2 mg of the soluble cell extract and samples were incubated for 2 h with end-over-end rotation at 4 °C. For GFP Immunoprecipitation, 1 mg of cell extract was incubated with 25 μL/sample of binding control magnetic agarose beads (Chromotek, bmab-20) for 40 min as a control of unspecific binding of proteins to the beads. Using a magnet, beads were precipitated, and the supernatant was transferred into a fresh tube containing 25 μL/sample of equilibrated GFP-trap magnetic agarose beads (Chromotek, gtma-20) and incubated for 1 h with end-over-end rotation at 4 °C. For Flag immuno-precipitation, between 100 and 800 μg of cell extract was incubated with 40 μL/sample of Anti-FLAG® M2 affinity gel (Sigma, A2220) for 2 h with end-over-end rotation at 4 °C. Immunoglobulin-antigen complexes were washed 3 × 15 min in cold PBS with end-over-end rotation before elution in 2× SDS sample buffer, except for Flag immunoprecipitation, where immunoglobulin antigen complexes were washed 3 × 15 min in cold TBS (50 mM Tris-HCl pH7.4, 150 mM NaCl) with end-over-end rotation before elution in 2x SDS sample buffer. For HA-immunoprecipitation to detect interaction between purified recombinant proteins, 800 μg of CHO protein extracts containing the bait protein (HA-tagged TOPBP1 aa740-899) were incubated with 30 μL of monoclonal anti-HA agarose beads for 1 h at 4 °C end-over-end rotation. After incubation, the unbound fraction was

washed away and 5 μg of purified GST-tagged ArmRD was added, followed by incubation for 90 min at 4 °C, end-over-end rotation. Washing steps were performed (3 × 15′ with cold PBS) to remove the excess of unbound GST-tagged ArmRD. After the last wash, protein complexes were eluted using 50 μL of 2x SDS sample buffer.

**Mass spectrometry**. The Functional Genomics Center of the University of Zurich (FGCZ) was commissioned to perform the mass spectrometry analysis. HA immunoprecipitation was performed as described above. After the last wash, precipitated material was resuspended in PBS buffer pH 7.45 and subjected to on-beads trypsin digestion according to the following protocol. Beads were washed twice with digestion buffer (10 mM Tris, 2 mM CaCl₂, pH 8.2). After the last wash, the buffer was discarded and monoclonal anti-HA agarose beads were resuspended in 10 mM Tris, 2 mM CaCl₂, pH 8.2 buffer supplemented with trypsin (100 ng/μL in 10 mM HCl). The pH was adjusted to 8.2 by the addition of 1 M Tris pH 8.2. Digestion was performed at 60 °C for 30 min. Supernatants were collected, and peptides were extracted from beads using 150 μL of 0.1% trifluoroacetic acid (TFA). Digested samples were dried and reconstituted in 20 μL ddH2O + 0.1% formic acid before performing liquid chromatography-mass spectrometry analysis (LC MS/MS). For the analysis 1 μL were injected on a nanoACQUITY UPLC system coupled to a Q-Exactive mass spectrometer (Thermo Scientific). MS data were processed for identification using the Mascot search engine (Matrixscience) and the spectra was searched against the Swissprot protein database.

**Immunofluorescence**. Cells were grown on glass coverslips (Thermo Scientific Menzel) and washed two times with cold PBS before fixation with cold methanol for 10 min on ice. Methanol was discarded and cells were washed 2 × 5 min with PBS at room temperature and incubated with blocking buffer (10% FCS in PBS) for at least 1 hour. Primary antibody incubations were performed at 4 °C overnight. Coverslips were then washed 3 × 10 min with 10% FCS in PBS and secondary antibody incubation was performed for 1 h at room temperature in the dark. After washing 3 × 10 min with PBS, coverslips were mounted on glass microscopy slides (Thermo Scientific) with VECTASHIELD® PLUS Antifade Mounting Medium containing 0.5 μg/mL 4′,6-diamidino-2-phenylindole dihydrochloride (DAPI; Vector Laboratories). The following antibodies were used at the indicated dilutions: CIP2A (Mouse, Santa Cruz, sc-80659, 1:800), γH2AX (Mouse, Merck, 05-636-I, 1:500), MDC1 (Rabbit, Abcam, ab11171, 1:300), TOPBP1 (Rabbit, Millipore, ABE1463, 1:300) CENPA (Mouse, Abcam, ab13939, 1:500), pSer10 Histone H3 (Rabbit, Millipore, 06-570, 1:500) ACA (Human, Antibodies Incorporated, 15–235, 1:2000), Cep135 (rabbit, home-made, 1:1000)[36]. The following secondary antibodies were used: Alexa Fluor 488 Goat Anti-Mouse (Life Technologies, A11029, 1:500), Alexa Fluor 568 Goat Anti-Mouse (Life Technologies A11031, 1:500), Alexa Fluor 647 Goat Anti-Mouse (Life Technologies A21235, 1:500), Alexa Fluor 488 Goat Anti-Rabbit (Life Technologies A11034, 1:500) and Alexa Fluor 568 Goat Anti-Rabbit (Life Technologies, A11036, 1:500).

**In-situ proximiy-ligation assay (PLA)**. U2OS cells were seeded on round glass coverslips in 24-well plates at a density of $0.8 \times 10^5$ cells per well. The growth medium was removed and all coverslips were washed with PBS, fixed with ice-cold methanol on ice for 12 min, and washed again 3x in PBS. All subsequent steps were performed using Duolink® PLA Reagents (Sigma-Aldrich, DUO92001, DUO92005, DUO92008, DUO82049). The blocking solution was applied for 1 h in the humidified incubator at 37 °C. The following antibody combinations were used: mouse-anti- CIP2A (Mouse, Santa Cruz, sc-80659, 1:800), and anti-TOPBP1 (Rabbit, Millipore, ABE1463, 1:300) or anti-MDC1 (Rabbit, Abcam, ab11171, 1:300). Antibody incubation was done overnight at 4 °C under humid conditions. PLUS and MINUS PLA probes were prepared according to the manufacturer's protocol and applied analogous to the previous step but for 1 h at 37 °C in the humidified incubator. Cells were washed for 15 min, then 10 min with wash buffer B, followed by 5 min with 0.01X wash buffer B, and mounted on glass slides using VECTA-SHIELD® PLUS antifade mounting medium with DAPI.

**Widefield and confocal microscopy**. Widefield image acquisition was done on a Leica DMI6000B inverted fluorescence microscope, equipped with Leica K5 sCMOS fluorescence camera (16-bit, 2048 × 2048 pixel, 4.2 MP) and Las X software version 3.7.2.22383. An HC Plan Apochromat 40×/0.95 phase contrast air objective and HCX Plan Apochromat 63X/1.40, and Plan Apochromat 100X/1.40 phase contrast oil immersion objectives were used for image acquisition. For triple-wavelength emission detection, we combined DAPI with EGFP or Alexa Fluor 488 and Alexa Fluor 568.

Confocal images were acquired with a Leica SP8 confocal laser scanning microscope coupled to a Leica DMI6000B inverted stand, with a 63x, 1.4-NA Plan-Apochromat oil-immersion objective. For quadruple-wavelength emission detection we combined DAPI with EGFP, Alexa Fluor 568 and Alexa Fluor 647. For triple-wavelength emission detection we combined DAPI with EGFP or Alexa Fluor 488, and Alexa Fluor 568. The sequential scanning mode was applied, and the number of overexposed pixels was kept at a minimum. 7 to 10 z-sections were recorded with optimal distances based on Nyquist criterion.

For airyscan high-resolution confocal microscopy cells were grown on high precision glass coverslips #1.5H, 0.17 mm thick (Assistent) and processed for immunofluorescence as described above. Cells were then stained with 1 µg/mL DAPI (AppliChem, A1001) diluted in PBS for 15 min at room temperature in the dark. DAPI was discarded and cells were further washed 2 × 5 min with deionized water to completely remove the PBS. Coverslips were mounted on microscopy glass slides without frosted edges (R. Langenbrick GmbH, Dimensions L 76 × W 26 mm) using VECTASHIELD® HardSet™ Antifade Mounting Medium (Vector Laboratories). Airyscan confocal imaging was carried out at the ScopeM imaging facility of ETH Zurich, using an LSM 880 Airyscan inverted microscope (ZEISS) equipped with a DIC Plan Apochromat 63×/1.4 oil immersion objective and an Airyscan 32-pinhole detector unit. DAPI was detected using a 405-nm diode laser and 420–480 nm plus 495–550 nm band pass emission filters, Alexa 488 and GFP were detected with the 488-nm line of an argon laser and 420–480 nm plus 495–550 nm band pass emission filters, and Alexa 568 was detected using a 561-nm diode-pumped solid-state laser and 495-550 nm band pass plus 570 nm long pass emission filters. Z-stacks of entire cells were acquired using optimal step size. Raw data were processed using Airyscan processing with Wiener Filter (auto setting) available in ZEN Black software version 2.1, yielding 8-bit images with approximately 180 nm lateral resolution. Display of images was adjusted for intensity for optimal display of structures of interest.

For optimal representation in figures, maximum intensity projections were calculated using Fiji[37]. Unprocessed grayscale tagged image files (TIFs) and maximum intensity projections of confocal z-stacks were exported from Fiji, followed by pseudo-coloring and adjustment of exposure or brightness/contrast in Adobe Photoshop CC2021. For maximum data transparency and preservation, unprocessed grayscale images or maximum intensity projections were imported as smart objects and adjustments were done using adjustment layers. Processed images were saved as multilayer PSD files.

**Image quantification**. DNA damage foci in mitotic cells were counted manually. For quantitative assessment of protein colocalization, the SQUASSH plugin (part of the MosaicSuite) for ImageJ and Fiji was used[11,38]. PLA quantification and TOPBP1 foci quantification in interphase cells were done using CellProfiler 4.0[39]. First, nuclei segmentation was performed by the intensity-based primary object detection module using the DAPI signal. For DNA damage foci segmentation, the primary object detection module was used on the respective channels after applying a feature enhancement filter. Downstream data manipulation was done using either RStudio or structured query language (SQL) scripts on the CellProfiler output SQLite databases. CellProfiler pipelines and SQL scripts are available upon request. For quantification of γH2AX foci, cells were grown on coverslips and incubated with RO-3306 for 16 h to synchronize cell population in late G2. Cells were subsequently released from the cell cycle arrest by washing 3 × 5 min with warm PBS and by adding fresh culture medium. Cells were further incubated for 20 min to allow them to progress into mitosis before irradiation with 0.5 Gy, when indicated, and then fixed 24 h after irradiation with ice-cold methanol. Cells were stained with mouse monoclonal anti-γH2AX antibody as described above. Image acquisition was done using widefield microscopy and quantification was done using CellProfiler 4.0 as described above. For quantification of micronuclei formation, unsynchronized cell populations were grown on coverslips and either treated with 3 Gy or left untreated. 6 h after IR cells were washed in cold PBS and fixed with ice-cold methanol for 10 min. Methanol was discarded and cells were washed twice before mounting the coverslips on microscopy glass slides with VECTASHIELD® PLUS antifade mounting medium with DAPI. Images were captured by widefield microscopy and micronuclei were counted manually in Fiji, while nuclei were counted using a CellProfiler 4.0 nuclei segmentation pipeline. For quantification of CENPA and γH2AX positive and negative MNi, cells were stained with anti-CENPA or anti-γH2AX antibodies, following standard immunofluorescence protocol, as described above. Image acquisition was done using widefield microscopy and quantification of CENPA and γH2AX positive and negative MNi was done manually.

**Metaphase analysis**. Cells were grown to 90% confluence on a 10 cm plate and were treated with 0.1 µg/mL KaryoMax Colcemid (Thermo Fisher Scientific) for 16 h overnight. Cells were trypsinized and transferred to a 15 ml Falcon™ tube, centrifuged at 176 g for 5 min and carefully resuspended in 5 ml of pre-warmed hypotonic buffer (15% FBS, 75 mM KCl) with intermittent agitation and incubated for 15 min at 37 °C. Cells were again pelleted at 176 g for 5 min, the supernatant was discarded, and the cell pellet was resuspended in 200 µl of hypotonic buffer. Cells were fixed by adding drop-wise 7 ml of ice-cold MeOH:AcOH 3:1 while slowly vortexing followed by 40 min on ice. After centrifugation at 176 g for 5 min, supernatant was discarded, and cells were resuspended in the remaining 200 µl of the fixation buffer. A total of 20–25 µl of the cell suspension was then dropped at a 45° angle onto a wet glass slide and air-dried for 10 min. Metaphases were stained with VECTASHIELD® PLUS antifade mounting medium with DAPI, covered with a glass coverslip and sealed with nail varnish. Telomere FISH was conducted using the Telomere PNA FISH Kit/Cy3 (Dako, K5326) according to the protocol provided by the supplier. Spreads were imaged using Leica DMI6000B inverted fluorescence microscope and quantification of telomere aberrations was performed manually.

**Clonogenic survival assay**. For clonogenic survival assay, $4 \times 10^5$ asynchronously growing RPE-1 cells were counted and plated in 6 cm dishes. The day after plating, cells were treated with the indicated doses of X-rays. Soon after irradiation, cells were counted and plated in triplicates in 6-well plates, at a density of 250 cells/well. Colonies were grown for 14 days at standard cell culture conditions with medium change every two days. At the assay endpoint, colonies were fixed with 3.7% formaldehyde (Chemie Brunschwig, extra-pure 37-41% solution diluted 1/10 in deionized water) for 10 min before staining with 0.5% Crystal Violet in 10% Ethanol solution (Honeywell) for 10 min at RT. Last, staining solution was discarded, colonies were washed 3 × 2 min in PBS while shaking and left to dry overnight. Images of the plates were acquired using a flatbed scanner Epson perfection V800 (Epson, Nagano – Japan) with the following settings: 24-bit colour, 1200 dpi resolution. Quantification was performed using the ColonyArea plug-in of Fiji[40].

**Statistics and reproducibility**. The appropriate statistical test was chosen as follows: Unpaired normal distributed data were tested with a two-tailed t test (in case of similar variances) or with a two-tailed t test with Welch's correction (in case of different variances). Unpaired non-continuous (count) data or non-normal distributed data were tested with two-tailed Mann–Whitney test (in case of similar variances) or with a two-tailed Kolmogorov–Smirnov test (in case of different variances). Three or more groups were analyzed by one-way ANOVA with Dunnett's correction for multiple comparison. In case of non-continuous (count) data or non-normal distribution, Kruskal-Wallis with Dunn's correction for multiple comparison was used. Clonogenic survival data was tested with linear regression.

All statistical analysis was performed with GraphPad Prism 9. Sample sizes and the statistical tests used are specified in the figure legends. Micrographs (Figs. 1b; 2a–c; 3b, e; 4b; 5a, e and Supplementary Figs. 2e; 3b–c, e; 5b; 8c–d; 9a–b) show representative examples from experiments that were repeated independently at least 3 times with similar results. Micrographs in Figs. 2e; 7a, c, f; 8a; 9f and Supplementary Figs. 2c; 5d; 7c; 8a; 9c show representative examples from experiments that were repeated independently at least 2 times with similar results. Western blots in Figs. 1e; 3d; 4a show representative examples from experiments that were repeated independently at least 3 times with similar results. Western blots in Figs. 1c, d; 3a; 5c; 6a–b; 7e and Supplementary Figs. 3d; 4b–c; 7b show representative examples from experiments that were repeated independently at least 2 times with similar results. The control experiments shown in Supplementary Figs. 2a–b, d; 5a, c, e and 6 were performed once.

**Reporting summary**. Further information on research design is available in the Nature Research Reporting Summary linked to this article.

## Data availability

The mass spectrometry proteomics data generated in this study have been deposited in the ProteomeXchange Consortium via the PRIDE[41] partner repository under accession code identifier PXD034100. Primary imaging data are available in the BioImage Archive under accession number S-BIAD492. Source data are provided with this paper.

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

## Acknowledgements

We thank Thanos Halazonetis, Steve Jackson, Graham Dellaire, Brian McStay and Roger Greenberg for providing valuable reagents. Imaging was performed with equipment maintained by the Center for Microscopy and Image Analysis, University of Zürich or the Scientific Center for Optical and Electron Microscopy (ScopeM) of the ETH Zürich. Cell sorting was carried out by the Flow Cytometry Core Facilities at the University of Zürich. Mass spectrometry was carried out by the Functional Genomics Center of the University of Zürich. The Blackford lab is supported by a CRUK Career Development Fellowship (C29215/A20772) and an Against Breast Cancer/Oriel College Research Fellowship to A.N.B., and an MRC WIMM DPhil Prize Studentship (MR/N013468/1) to A.-M.K.S. The Durocher lab is supported by a grant from the CIHR (FDN143343). The Stucki lab is supported by project grants from the Swiss National Foundation (31003A_163141 and 310030_189141), the Groeber Foundation and by the Kanton of Zürich.

## Author contributions

The project was conceived and supervised by M.S. and D.D. Biochemical and cell biological experiments were carried out and data were analysed by M.D.Z., M.T.E., C.M., S.A., S.E.R., A.J., P.-A.L., D.F. and M.S. A.-M.K.S. generated reagents. D.D. supervised S.A. and S.E.R. A.N.B. supervised A.-M.K.S. The paper was written by M.S. with contributions from other authors.

## Competing interests

D.D. is a shareholder and advisor of Repare Therapeutics. The remaining authors declare no competing interests.
