## [Peer Review File · Nature Communications]

The CIP2A-TOPBP1 complex safeguards chromosomal stability during mitosisREVIEWER COMMENTS

Reviewer #1 (Remarks to the Author):

De Marco Zompit et al present data within the manuscript identifying CIP2A as a novel TOPBP1 interacting protein that is specifically required for TOPBP1 to form IRIF in mitosis, as it is normally excluded from the nucleus in interphase due to an NES.

Overall the manuscript is well written and the data presented looks convincing. However, I have one major concern with this manuscript and that the analysis presented is extremely superficial and majorly lacks any mechanistic insight as to how CIP2A really contributes to maintaining genome stability. Demonstrating that CIP2A is required for TOPBP1 to form foci in mitosis doesn't really cut it for me in terms of demonstrating mechanism. Whilst the manuscript is superficially interesting, it does not come anywhere close to being ready for publication in Nature Communications. To me, the paper seems a little rushed out, presumably because the Durocher lab has their own manuscript on CIP2A on BioRxiv. Lastly, this manuscript suffering from a distinct lack of proper quantification throughout. None of the PLA images are quantified. Many of the images quantified by SQUASSH have an 'n' number of 10 or less. This is not acceptable. PLA quantification should be carried out on at least 100 cells per experimental repeat. Whilst I am not familiar with SQUASSH, personally I would not believe any data presented that has been quantified from 6 cells.

Specific points:

1. Figure 2d n=6. Needs significantly more quantification.
2. Figure 2f. No quantification. No siRNA depletion of either CIP2A or TOPBP1 as control for non-specific PLA spots.
3. Figure 3c n=10. Needs significantly more quantification.
4. Figure 3d. No quantification.
5. Figure 3e. No quantification and no Western blot demonstrating the level of WT and mutant TOPBP1 expression.
6. Figure 4b n=16. Needs significantly more quantification.
7. Figure 4c. Why has the MDC1 phospho-specific mutant that lacks the ability to bind TOPBP1 not been used as a control?
8. Figure 4e. No quantification. No siRNA depletion of either CIP2A or MDC1 as control for non-specific PLA spots.
9. Figure 4g n=9. Needs significantly more quantification.
10. Figure 5b/c. The quantification +/- leptomycin B goes from 4 to 4.3 respectively. Firstly, the quantification does not match the images and secondly, how is an increase of 0.3 biologically relevant?
11. Figure 5h. No quantification. No siRNA depletion of either CIP2A or TOPBP1 as control for non-specific PLA spots.
12. Figure 7a. Epistasis should be carried out with MDC1, TOPBP1 and CIP2A for MN.
13. Figure 7b. Are the CENPA negative MN positive/negative for DSBs e.g. 53BP1 foci?
14. Figure 7c/d: What are the aberrations quantified? From the image it looks as though the primary defect in cells lacking CIP2A is relative to dysfunctional telomeres. Again no epistasis has been carried out with MDC1, TOPBP1 and CIP2A for chromosome aberrations.
15. It is not clear to me what CIP2A is doing to prevent genome instability. The authors have not conclusively shown that any genome instability in cells lacking CIP2A results from a defect repair mitotic DSBs.
16. Does loss of CIP2A increase genome instability under situations of increased replication where cells are still undergoing DNA synthesis in mitosis?

Reviewer #2 (Remarks to the Author):

In their manuscript, De Marco Zompit et al. describe that the protein CIP2A interacts with TOPBP1, and colocalizes with TOPBP1 in IR-induced foci, specifically in mitotic cells. They subsequently show that CIP2A recruitment to these foci is dependent on TOPBP1 expression (and vice versa), and on expression of MDC1 with intact S168 and S196 phospho-acceptor sites. In addition, their data indicate that CIP2A recruitment to TOPBP1 foci is specific to mitotic cells because CIP2A localizes in the cytoplasm in interphase cells, and can only enter the nucleus upon nuclear envelope breakdown at the onset of mitosis. Finally, the authors show that CIP2A-deficient cells have a reduced capacity to repair mitotic DSBs, and display a genomic instability phenotype.

The manuscript is well written and the data are clearly displayed. The finding that CIP2A is involved in mitotic DSB-repair is novel and relevant, and the manuscript presents some mechanistic understanding of CIP2A's function in this process. In most cases, the conclusions are well supported by the data. I do however have a several comments and suggestions for experiments to further improve the manuscript and substantiate the conclusions. Provided these concerns are adequately addressed, the manuscript may be considered for publication.

Major Comments

- The authors performed IP-studies (Fig. 1C, D; Fig. 4C, D) and IF-experiments (Fig. 2) to convincingly show that CIP2A and TOPBP1 interact, specifically in mitotic cells (Fig. 2, Fig. 5H). However, this interaction is not specific to irradiated cells, as can be observed in the PLA experiments (Fig. 2F and particularly Fig. 5H), and is apparent from the fact that the IP-experiments are performed with extracts from untreated cells. To better analyze the CIP2A and TOPBP1 interaction, IP-studies should be performed in untreated and IR-treated cells that are either asynchronous or synchronized in mitosis.
- In untreated mitotic cells, the PLA-results also seem to contradict the IF-studies, as the first shows evenly distributed foci whereas the latter shows two centrosome-associated punctae (compare Fig. 2A to 5H). This should be more thoroughly interpreted and explicitly discussed.
- The authors perform MS-studies and an initial validation IP (Fig. 1C) with TOPBP1 region 740-899, and show that this region is sufficient to mediate CIP2A interaction. However, the TOPBP1 deletion mutant they subsequently test for CIP2A interaction misses a smaller region, namely aa 813-892. The authors argue that this is the conserved region, but region 790-810 is also conserved, and even 892-899 contains conserved residues (Fig. S1A). This is important because the TOPBP1 D813-892 mutant still interacts with CIP2A, albeit slightly reduced (Fig. 1D). The authors should thus more carefully map the CIP2A-interacting region in TOPBP1, and at least also study a D790-899 mutant.
- Related to the previous point, the blot shown in Fig. 1D is not very convincing. Thus, to substantiate the reduced interaction between CIP2A and the TOPBP1 D813-892 mutant, the authors should reverse the IP and study TOPBP1 WT and mutant pulldown after precipitating CIP2A (either endogenous or tagged).
- The TOPBP1 siRNA results in reduced CIP2A expression levels (Fig. S2D), and the authors therefore state in line 142 that it is not possible to determine on basis of the data presented in Fig. 3D whether loss of the TOPBP1-CIP2A and/or reduced CIP2A expression affects CIP2A recruitment to IR-foci. This could potentially be resolved by performing TOPBP1 RNAi in the +FLAG-CIP2A cells, which strongly overexpress CIP2A (Fig. 3A).

- Related to the previous point, the TOPBP1 D813-892 mutant shows a mildly reduced interaction with CIP2A (Fig. 1D), and is not recruited to IR-foci (Fig. 1B). The lack of CIP2A localization to IR-foci, shown in Fig. 3E, could therefore be explained by either a requirement for TOPBP1 localization at the DSB-site, or a requirement for interaction of CIP2A with TOPBP1 (Fig. 3E). It is thus not possible to conclude, as the authors do in line 151, that CIP2A recruitment is dependent on its interaction with TOPBP1. This emphasizes again that the authors should map the interaction domain in more detail and repeat these experiments with a TOPBP1 mutant that fails to interact with CIP2A.
- How can TOPBP1 foci be dependent on CIP2A (see Figure 4A), since TOPBP1 expression is not affected and TOPBP1 can still interact with MDC1 following CIP2A loss (see Supplementary figure 2D and Figure 4H, respectively)? The authors should clarify this experimentally to provide mechanistic insight into how CIP2A contributes to TOPBP1 foci formation.
- Based on their findings in Figure 5, the authors claim that CIP2A is a direct CRM1 ligand. Although the available mass spectrometry interaction data may support this, the authors should verify if CRM1 and CIP2A interact in a manner dependent on its NES at 598-612.
- The authors conclude that CIP2A and TOPBP1 interact at sites of DSBs in mitosis (line 113). However, they formally only show CIP2A/TOPBP1 colocalization in foci on DNA. The authors should at least repeat the experiments shown in Fig. 2A, B and include a γ H2AX stain to show that the CIP2A/TOPBP1 foci colocalize with sites of damage. Furthermore, IR can induce many kinds of damage, not only DSBs. To formally conclude the colocalization occurs at DSBs, the authors should do this after inducing bona-fide DSBs using nuclease systems such as AsiSI, Cas9 or FokI.
- How CIP2A mechanistically contribute to DSB repair and genome stability maintenance remain rather unclear. To address this further the authors should show whether CIP2A contributes to TOPBP1-dependent bridging of MDC1-Bound DSBs Acquired during Mitosis as in their previous work, Pia Amata-Leimbacher et al., Mol. Cell (2019).

Minor Comments

- "Inspection of one of these DNA damage chemogenetic datasets revealed a significant drug sensitivity correlation between MDC1 and CIP2A (Supplementary Fig. 1d)." How do the authors know this is a significant correlation? No numbers/statistics are provided to warrant this conclusion.
- In Fig. 3B, to be able to fully appreciate the extent of colocalization, the authors should include pictures of the individual channels, like they do for the other IF-panels.
- Whereas all IF-pictures demonstrating colocalization between CIP2A and TOPBP1 are of cells in prophase, the siTOPBP1 example cell shown in Fig. 3D has clearly progressed to pro-metaphase and appears close to metaphase. For a fairer comparison of the siTOPBP1 condition to the other data, the authors should show a cell that is in prophase, like the other example cells shown.
- A western blot associated to figure 3E, showing the expression of the TOPBP1 variants, is missing.
- Also on Figure 3e, was the expression of CIP2A affected in cells expressing TOPBP1 Δ 813-192? If so, reduced CIP2A foci formation in these cells may also stem (partially) from this effect. This can be clarified by Western blot analysis (as in Supplementary figure 2d).
- In line 110, line 148 and line 149 the TOPBP1 D813-892 mutant is inadvertently written as D813-192 mutant.
- Figures 1F and 4E, did secondary antibody controls only show no signals in the PLA assay?

- Figure 3e, was the expression of CIP2A affected in cells expressing TOPBP1 Δ 813-192? If so, reduced CIP2A foci formation in these cells may also stem (partially) from this effect. This can be clarified by Western blot analysis (as in Supplementary figure 2d).
- A western blot is missing to validate (loss of) MDC1 expression in the DMDC1 cells with and without GFP-MDC1 WT or mutant (Fig. 4).
- To clarify figure 7A, the figure legend should include what each symbol represents, in addition to explaining this in the materials and methods section.

Reviewer #3 (Remarks to the Author):

Review comments

of "The CIP2A-TOPBP1 complex safeguards chromosomal stability during mitosis"

This manuscript by Mara De Marco Zompit et al. describes a study which identified CIP2A as a TOPBP1-MDC1 interacting protein that promotes TOPBP1 recruitment to sites of mitotic DSBs in mitosis. Cells lacking CIP2A display increased micronuclei formation, DSB repair defects, and chromosomal instability. CIP2A is actively exported from the cell nucleus in interphase but, upon nuclear envelope breakdown at the onset of mitosis, gains access to chromatin. Therefore, authors propose CIP2A-TOPBP1 as a mitosis-specific genome maintenance complex.

CIP2A is well-documented oncogene and a presumed therapeutic target for breast cancer. The protein is also a well-known PP2A inhibitor and was shown to stabilize MYC in human malignancies. The discovery of CIP2A's role in maintaining genome stability has implications on the use of CIP2A inhibitor in a clinical context and therefore is interesting and important. Mara De Marco Zompit et al. propose CIP2A as a TOPBP1-interacting protein that promotes TOPBP1 recruitment to sites of DSBs specifically in mitosis in MDC1-dependent manner. TOPBP1 is known to specifically interact with Casein kinase 2 (CK2)-phosphorylated MDC1 and this interaction is required for mitotic recruitment of TOPBP1 to DSB sites by MDC1. CIP2A is a well-established PP2A inhibitor, and the authors propose that CIP2A promotes TOPBP1 mitotic recruitment to DSB site by MDC1 without considering the possibility that CIP2A inhibits PP2A activity in dephosphorylating MDC1?

This manuscript by Mara De Marco Zompit et al. is carefully written, most experiments are well designed, and most data are clean. However, there is a consistent lack of proper controls and DNA damage/mitotic markers that negatively impacts the quality of the manuscript. Figures are not labeled clearly and are not presented in a way that is easy to understand. The lack of statistical (?) analysis on about half of all figures and the missing mechanistic connection between CIP2A-TOPBP1 interaction and increased micronuclei to cell death leave the story half finished. Before reaching their conclusion that CIP2A binds to TOPBP1-MDC1 complex to promote TOPBP1 recruitment to sites of mitotic DSBs to promote genome stability, there are some main issues that need to be addressed:

Major concern:

1. Figure 1. To demonstrate that CIP2A directly interacts with TOPBP1, endogenous IP and in vitro binding experiments with recombinant protein are needed. Also, the authors need to demonstrate whether CIP2A's ability to inhibit PP2A is required for the interaction between CIP2A and TOPBP1 and between TOPBP1 and MDC1.
2. Does the TOPBP1-CIP2A interaction increase after IR and over mitosis in IP-WB?

3. Figure 2. As shown in figure 2A, TOPBP1 and CIP2A also interact during mitosis in the absence of IR although in less foci. Authors need to determine if the TOPBP1-CIP2A interaction during mitosis is co-localized with gamma H2Ax foci, or centrosome and unattached kinetochore. Increased DNA damage can increase the number of unattached kinetochores or can delay kinetochore attachment to mitotic spindle, which can lead to a similar increase in mitotic foci formation.
4. Figure 3. Please demonstrate that loss of CIP2A or TOPBP1 does not lead to a corresponding change of TOPBP1 or CIP2A total protein level.
5. Figure 3d. Please demonstrate mitotic DSB foci number is not affected by loss of CIP2A or TOPBP1 with gamma-H2Ax foci.
6. Figure 4c. MW of GFP-MDC1 pulled down is higher than input.
7. Figure 4f. Does MDC1S168E/S196E mutant bring TOPBP1 to mitotic foci? Will CIP2A loss still impact TOPBP1 mitotic foci formation in the presence of MDC1S168E/S196E?
8. Figure 4h. Please include IP-WB in mitotic cells as Figure 5 indicates CIP2A does not colocalize with TOPBP1 during interphase.
9. Figure 5, 6. If 3XNLS tag is added to CIP2A-DeltaNES mutant, will that bring CIP2A to nucleus and regulate MDC1-TOPBP1 interaction after IR induced DNA damage?
10. Figure 6. Can you show interphase TOPBP1 foci with 1Gray IR?
11. Figure 7a. Please indicate how many cells were quantified. Also plot -IR and +IR in the same graph and perform statistical analysis over CIP2A-/- cells before and after IR.
12. Figure 7e, IR dose changed to 0.5 Gray again? Why? Please keep the IR dose consistent or show results in dosage and timepoint.

Minor issue:

1. Figure 1a. Please indicate BRCT domain on schematic for assigned number 0-8, also include aa sequence number on schematic.
2. Figure 1b. There is a lack of quantification and statistical analysis. Also, the authors need include a mitotic marker such as p-Histone H3 and a DNA damage marker such as gamma H2Ax.
3. Figure 1b. The Mitosis and Interphase [what] use different doses of IR? Please use same dose to exclude the possibility that the difference is due to the difference in IR dose.
4. Figure 2a, 2b. The Mitosis and Interphase [what] use different doses of IR? Please use same dose to exclude the possibility that the difference is due to the difference in IR dose.
5. Figure 2. There is a lack of quantification and statistical analysis. Also, the authors need include a mitotic marker such as gamma tubulin or mad2 and a DNA damage marker such as gamma H2Ax.
6. All figure with images, label the scale bar on image.
7. Figure 2D. there is a lack of a mitotic marker like p-Histone H3.

8. Figure 3B. Include MAD2 and α -tubulin staining as localization control for centromere and centrosome (mitotic spindle).

9. CIP2A and TOPBP1 are both required to recruit one another to mitotic foci, the title in result section did not reflect the fact that TOPBP1 is also required for CIP2A recruitment to mitotic foci and therefore is misleading.

10. Figure 6b, statistical analysis needed.

11. Figure 7c, please clearly label on the figure what the red dots indicating.

12. Figure 7d, statistical analysis needed.

13. Figure 7e, please label clearly on figure what the red dots indicating or use red font to indicate to be consistent with green γ H2Ax and blue DAPI.

Response to Reviewers

1. Overview

We thank the reviewers for agreeing to evaluate our work and for their detailed and insightful comments. These have been invaluable for us to further strengthen our manuscript and have guided our revisions, which include addition of a sizable amount of new data.

Below, we provide a detailed point-by-point response to all specific questions raised by the reviewers. Reviewers' comments are italicized.

2. Point-by-point response

Reviewer #1:

De Marco Zompit et al present data within the manuscript identifying CIP2A as a novel TOPBP1 interacting protein that is specifically required for TOPBP1 to form IRIF in mitosis, as it is normal excluded from the nucleus in interphase due to an NES.

Overall the manuscript is well written and the data presented looks convincing. However, I have one major concern with this manuscript and that the analysis presented is extremely superficial and majorly lacks any mechanistic insight as to how CIP2A really contributes to maintaining genome stability. Demonstrating that CIP2A is required for TOPBP1 to form foci in mitosis doesn't really cut it for me in terms of demonstrating mechanism. Whilst the manuscript is superficially interesting, it does not come anywhere close to being ready for publication in Nature Communications. To me, the paper seems a little rushed out, presumably because the Durocher lab has their own manuscript on CIP2A on BioRxiv. Lastly, this manuscript suffering from a distinct lack of proper quantification throughout. None of the PLA images are quantified. Many of the images quantified by SQUASSH have an 'n' number of 10 or less. This is not acceptable. PLA quantification should be carried out on at least 100 cells per experimental repeat. Whilst I am not familiar with SQUASSH, personally I would not believe any data presented that has been quantified from 6 cells.

We thank the Reviewer for this overall evaluation and for stating that our manuscript is well written and that the data look convincing. The main criticism of this Reviewer focuses on the lack of quantification throughout. More specifically, some experiments lacked quantification completely and some did include quantification, but in the opinion of this Reviewer, the sample size (n) was in some instances too small. We have now included quantification for all the main experiments. In addition, we significantly increased the sample size in all the quantifications and in most cases, pooled the data from several independent experiments to

remove experimentation bias. Having said this, we would like to point out that we disagree with this Reviewer's opinion that 'at least 100 cells' need to be assessed per experimental repeat. As most of the mitotic analysis had to be done on a (slow) confocal microscope (point scanner) at high magnification, imaging 100 cells per experiment is an impossible task. The minimal sample size depends on the effect size and thus, the statistical power. It can be calculated for each experiment. Since the effect size in most of our experiments was huge, there is no need to have a large sample size of 100 cells. On the contrary, increasing the sample size also increases the statistical power, thus making even very small (and perhaps biologically irrelevant) differences between the groups statistically significant. Therefore, we believe that determining the minimal sample size for each experiment should be based on statistical considerations.

Specific points:

1. Figure 2d n=6. Needs significantly more quantification.

We significantly increased the sample size in this experiment (n=37) and pooled the data from three independent experiments to compensate for variation in the experimentation.

2. Figure 2f. No quantification. No siRNA depletion of either CIP2A or TOPBP1 as control for non-specific PLA spots.

A quantification of this experiment is now included and TOPBP1 was depleted by siRNA to control for the specificity of the signal (new Fig. 2f).

3. Figure 3c n=10. Needs significantly more quantification.

We significantly increased the sample size in this experiment (n=57 and 62) and pooled the data from three independent experiments.

4. Figure 3d. No quantification.

We completely redesigned this experiment (see also below in the comments of Reviewer #2) and we included a quantification and pooled the data from three independent experiments (new Fig. 3f).

5. Figure 3e. No quantification and no Western blot demonstrating the level of WT and mutant TOPBP1 expression.

We completely redesigned this experiment and included several additional TOPBP1 deletion mutants (see also below in the comments of Reviewer #2). Moreover, we added a quantification and Western blots to demonstrate the levels of WT and mutant TOPBP1 expression (new Fig. 4b,c and Supplementary Fig. 4b).

6. Figure 4b n=16. Needs significantly more quantification.

We significantly increased the sample size in this experiment (n=51-80) and pooled the data from three independent experiments (new Fig. 5b).

7. Figure 4c. Why has the MDC1 phospho-specific mutant that lacks the ability to bind TOPBP1 not been used as a control?

We repeated this experiment, this time using the MDC1 S168A/S196A mutant as a control (new Fig. 5c).

8. Figure 4e. No quantification. No siRNA depletion of either CIP2A or MDC1 as control for non-specific PLA spots.

A quantification of this experiment is now included, and we used MDC1 knock-out cells as a control for the specificity of the signal (new Fig. 5d).

9. Figure 4g n=9. Needs significantly more quantification.

We significantly increased the sample size in this experiment (n=21-24) and pooled the data from three independent experiments (new Fig. 5f).

10. Figure 5b/c. The quantification +/- leptomycin B goes from 4 to 4.3 respectively. Firstly, the quantification does not match the images and secondly, how is an increase of 0.3 biologically relevant?

In the label of the y-axis of this graph it was indicated that the values are the log of fluorescence intensity. Fluorescence intensities are often log-normally distributed due to lower sensitivity of the detector at low signal intensity. Log-transforming the values often remedies this issue, which allows parametric statistical tests to be performed on the data. An increase from log 4 to log 4.3 therefore means an actual increase in fluorescence intensity from 10^4 to $10^{4.3}$ = from 10,000 to ~20,000, which essentially means double the values on a non-logarithmic scale. This not only matches the images but is also clearly biologically relevant. However, for the sake of clarity, and since we noticed that the fluorescence intensities in this experiment were normally distributed, we revised the graph and displayed the data on a non-logarithmic scale (new Fig. 6b).

11. Figure 5h. No quantification. No siRNA depletion of either CIP2A or TOPBP1 as control for non-specific PLA spots.

A quantification of this experiment is now included, and TOPBP1 was depleted by siRNA to control for the specificity of the signal (new Fig. 6g).

12. Figure 7a. Epistasis should be carried out with MDC1, TOPBP1 and CIP2A for MN.

We performed an “epistasis” analysis by downregulating MDC1, TOPBP1 and CIP2A in control cells and MDC1 and CIP2A knock-out cells (see Figure below). The data indicate that

downregulation of TOPBP1 slightly increases MNi formation in CIP2A knock-out cells, which may suggest a synergistic interaction between these two genes. There was no significant increase of MNi in MDC1 knock-out cells in which CIP2A or TOPBP1 were downregulated, which may suggest an epistatic interaction between MDC1 and CIP2A and MDC1 and TOPBP1, respectively. However, we doubt that this analysis significantly adds to the story. It is well established that increased DNA replication stress increases MNi formation. Since TOPBP1 has important roles also in interphase cells, especially during S-phase where it is implicated in origin of replication firing and ATR activation, it is not surprising to see an increase in MNi formation when TOPBP1 is depleted in a CIP2A knock-out background. In addition, we did not detect increased MNi formation in control cells and CIP2A knock-out cells in which MDC1 was downregulated by siRNA, even though we did measure significantly increased numbers of MNi in MDC1 knock-out cells. This indicates that the MDC1 siRNA is not sufficiently effective to trigger increased MNi formation. We therefore believe that *bona fide* epistasis analysis would require the generation of double knock-out cell lines, which in our opinion, lies beyond the scope of this study and is not even possible in the case of TOPBP1, which is an essential gene. We therefore opted not to include this analysis in the revised manuscript.

Figure: “Epistasis” analysis of MNi formation. RPE-1 parental cells (WT), RPE-1 CIP2A knock-out cells (Δ CIP2A) and RPE-1 MDC1 knock-out cells (Δ MDC1) were transfected with control siRNA (siCtrl) and siRNAs against CIP2A (siCIP2A), TOPBP1 (siTOPBP) and MDC1 (siMDC1) as indicated. MNi formation was assessed by microscopy of DAPI stained slides. Bars and error bars represent the mean and standard error of three independent experiments.

13. Figure 7b. Are the CENPA negative MN positive/negative for DSBs e.g. 53BP1 foci?

We have now performed this analysis, using γ H2AX as a marker for DSBs (new Fig. 7c)

14. Figure 7c/d: What are the aberrations quantified? From the image it looks as though the primary defect in cells lacking CIP2A is relative to dysfunctional telomeres. Again no epistasis has been carried out with MDC1, TOPBP1 and CIP2A for chromosome aberrations.

The aberrations quantified are indicated in Fig. 7e (lower panels) and are described in the Figure legend to Fig. 7e. They include single chromatid telomere loss, sister chromatid telomere loss, interstitial telomeres, telomere duplications and dicentric chromosomes. We do not exclude additional chromosomal instabilities such as chromosome and chromatid breaks and gross chromosomal aberrations such as chromothripsis that have been associated with increased MNI formation. Since they are difficult or even impossible to detect on DAPI stained metaphase spreads, we did not score them. Yet, telomere loss (=terminal deficiencies) and dicentric chromosomes are the type of lesions expected to occur frequently when cells are unable to properly segregate acentric chromosome fragments. We did not perform epistasis analysis on chromosome aberrations for the reasons described above.

15. It is not clear to me what CIP2A is doing to prevent genome instability. The authors have not conclusively that any genome instability in cells lacking CIP2A results from a defect repair mitotic DSBs.

We apologize that this issue was not more clearly explained in the original version of the manuscript. We now added a more comprehensive Discussion to clarify this issue. Briefly, we are not claiming that genome instability in cells lacking CIP2A results from defective repair of mitotic DSBs. In fact, there is strong evidence that 'classical' DSB repair by NHEJ or HR is inhibited during mitosis (recently reviewed in ref. 1). There is some limited evidence that an alternative end joining pathway may be active in early stages of mitosis^{2 3}, but whether CIP2A is implicated in this process remains to be established. The phenotype of CIP2A knock-out cells however strongly suggests an inability to properly segregate acentric chromosome fragments during mitosis. This does not necessarily reflect a repair defect though. It is also possible that CIP2A is implicated in a mechanism that tethers the acentric fragment to the chromosome from which it has broken off. Evidence for such a mechanism has been presented in *Drosophila*, but not yet in mammalian cells⁴. Further work will be needed to clarify these issues.

16. Does loss of CIP2A increased genome instability under situations of increased replication where cells are still undergoing DNA synthesis in mitosis?

This question is addressed in another study from our laboratories that recently appeared online in *Nature Cancer*⁷. However, in that study a potential role of CIP2A in mitotic DNA synthesis (MiDAS) was not addressed. Therefore, we did the experiment and while we observed partial overlap between CIP2A foci and sites of MiDAS on mitotic chromosomes, CIP2A deficient cells did not display decreased MiDAS. On the other hand, we observed increased MiDAS in cells in which TOPBP1 was downregulated. These data are somewhat in conflict with a published report by the Lisby lab that implicated TOPBP1 in the process of MiDAS upon mild replication stress⁶. However, our data clearly demonstrate that CIP2A is not involved in the process of MiDAS in situations of increased replication stress, at least in U2OS cells. Yet, given that our manuscript is mostly about cellular responses to DSBs in mitosis and since MiDAS is, to our knowledge, not involved in the repair of mitotic DSBs, we decided not to include these data in the revised version of the paper.

a**b**
Figure: The role of CIP2A-TOPBP1 complex in MiDAS. **a** Confocal micrographs (maximum intensity projections) of U2OS cells transfected with control siRNA (siLUC) or siRNA against CIP2A (siCIP2A) and TOPBP1 (siTOPBP1). Cells were stained for incorporated EdU, CIP2A and TOPBP1. **b** The numbers of EdU foci per mitotic cell were quantified using SQUASSH. Left graph: data points represent mitotic cells collected from one representative replicate of the experiment (n=10). Statistical significance was calculated using Kruskal–Wallis nonparametric test, followed by Dunn’s multiple-comparison. Bars and error bars represent mean and SD. Right graph: mean of three independent experiments.

Reviewer #2

In their manuscript, De Marco Zompit et al. describe that the protein CIP2A interacts with TOPBP1, and colocalizes with TOPBP1 in IR-induced foci, specifically in mitotic cells. They subsequently show that CIP2A recruitment to these foci is dependent on TOPBP1 expression (and vice versa), and on expression of MDC1 with intact S168 and S196 phospho-acceptor sites. In addition, their data indicate that CIP2A recruitment to TOPBP1 foci is specific to mitotic cells because CIP2A localizes in the cytoplasm in interphase cells, and can only enter the nucleus upon nuclear envelope breakdown at the onset of mitosis. Finally, the authors show that CIP2A-deficient cells have a reduced capacity to repair mitotic DSBs, and display a genomic instability phenotype.

The manuscript is well written and the data are clearly displayed. The finding that CIP2A is involved in mitotic DSB-repair is novel and relevant, and the manuscript presents some mechanistic understanding of CIP2A's function in this process. In most cases, the conclusions are well supported by the data. I do however have a several comments and suggestions for experiments to further improve the manuscript and substantiate the conclusions. Provided these concerns are adequately addressed, the manuscript may be considered for publication.

We thank the Reviewer for this overall evaluation and for his/her opinion that our manuscript is well written, the data clearly displayed and that our findings on CIP2A are novel and relevant. We would also like to thank this Reviewer for his/her helpful comments and constructive criticism. We have addressed all the comments and added a significant amount of additional data to the revised version of our manuscript.

Major Comments

- The authors performed IP-studies (Fig. 1C, D; Fig. 4C, D) and IF-experiments (Fig. 2) to convincingly show that CIP2A and TOPBP1 interact, specifically in mitotic cells (Fig. 2, Fig. 5H). However, this interaction is not specific to irradiated cells, as can be observed in the PLA experiments (Fig. 2F and particularly Fig. 5H), and is apparent from the fact that the IP-experiments are performed with extracts from untreated cells. To better analyze the CIP2A and TOPBP1 interaction, IP-studies should be performed in untreated and IR-treated cells that are either asynchronous or synchronized in mitosis.

We have performed this experiment and added the result to the revised version of our manuscript (new Fig. 1e). The data clearly show that IR does not affect CIP2A-TOPBP1 interaction. However, we observed increased interaction between CIP2A and TOPBP1 in Nocodazole arrested cells. These results are consistent with the PLA data that show increased interaction between CIP2A and TOPBP1 in mitotic cells (Fig. 6g), but not in cells treated with IR (Fig. 2f).

- In untreated mitotic cells, the PLA-results also seem to contradict the IF-studies, as the first shows evenly distributed foci whereas the latter shows two centrosome-associated punctae (compare Fig. 2A to 5H). This should be more thoroughly interpreted and explicitly discussed.

By IF, we consistently observed CIP2A and TOPBP1 localization to the centrosomes as well as in a few small punctuate structures within the mitotic chromosome mass in untreated mitotic cells (Fig. 2a). Based on the findings of another study that we recently published⁷, we hypothesize that these small structures correspond to sites of DNA replication stress. In the PLA assay, the small foci on the mitotic chromosomes appeared much bigger and brighter (most likely due to the amplification reaction that is part of the PLA assay). They can therefore not be distinguished anymore from centrosomes, especially since in our hands, the PLA assay did not allow co-staining with a centrosome marker. To clarify this issue, we added additional data showing that in the absence of exogenous DNA damage treatment, CIP2A staining in IF mostly co-localizes with the centrosome marker Cep135 (new Supplementary Fig. 2a)

- The authors perform MS-studies and an initial validation IP (Fig. 1C) with TOPBP1 region 740-899, and show that this region is sufficient to mediate CIP2A interaction. However, the TOPBP1 deletion mutant they subsequently test for CIP2A interaction misses a smaller region, namely aa 813-892. The authors argue that this is the conserved region, but region 790-810 is also conserved, and even 892-899 contains conserved residues (Fig. S1A). This is important because the TOPBP1 D813-892 mutant still interacts with CIP2A, albeit slightly reduced (Fig. 1D). The authors should thus more carefully map the CIP2A-interacting region in TOPBP1, and at least also study a D790-899 mutant.

We thank Reviewer #2 for raising this important issue. In the revised version of our manuscript, we completely redesigned the TOPBP1-CIP2A interaction analysis and mapped the CIP2A-interacting region in TOPBP1 much more carefully (new Fig. 4). First, in Fig. 1b, we used a larger deletion construct lacking the entire conserved region between BRCT5 and 6 (Δ 751-899). We only left a very short 10 non-conserved amino acid linker (aa741-751) between BRCT5 and 6, to prevent folding issues of the two BRCT domains. We also generated two additional deletion constructs, the first one lacking a short conserved patch upstream of the original deletion (Δ 774-798). Interestingly, deletion of this conserved motif completely abrogated the interaction between TOPBP1 and CIP2A, while deletion of the more downstream conserved region (Δ 813-892; the original deletion construct we used in the first version of the manuscript) only partially abrogated the interaction (new Fig 4a). Combining both deletions within the same construct also completely abrogated the interaction. We therefore conclude that optimal interaction with CIP2A depends on both conserved amino acid patches located between BRCT5 and 6 of TOPBP1. The new data may also explain why the Δ 813-892 deletion construct still partially interacts with CIP2A in our co-IP experiments: it still contains the short upstream conserved region that seems to be more essential for an efficient interaction than the larger downstream region under these experimental conditions.

Having said this, we would like to point out that there is a discrepancy between these interaction data and two recently published studies using the yeast two hybrid system to map the CIP2A interaction site in TOPBP1^{7,8}. In the yeast two hybrid assay, a short sequence fragment between aa 830-852 was mapped as the minimal CIP2A interacting region and mutation of three amino acids within this short fragment (F838, D839 and V840) completely abrogated interaction between CIP2A and TOPBP1. Moreover, despite residual interaction with CIP2A in our co-IP experiments, TOPBP1 Δ 813-892 was completely defective for the recruitment to sites of mitotic DSBs, while deletion of the shorter patch (Δ 774-789) was only partially defective (new Fig. 4b). We therefore believe that the residual interaction observed in the Δ 813-892 deletion mutant may at least partially be caused by protein over-expression in our co-IP experiments. We now discuss this issue in the new Discussion part of the manuscript. Careful biochemical interaction studies with purified proteins will be necessary to resolve this issue. However, we feel that this is beyond the scope of this study.

- Related to the previous point, the blot shown in Fig. 1D is not very convincing. Thus, to substantiate the reduced interaction between CIP2A and the TOPBP1 Δ 813-892 mutant, the authors should reverse the IP and study TOPBP1 WT and mutant pull-down after precipitating CIP2A (either endogenous or tagged).

As described above, we have now completely redesigned this experiment and we think the new data in Fig. 4 are now more convincing. We also performed a reverse IP with Flag-tagged CIP2A with almost identical outcome (see Figure below). In this case the Δ 813-892 mutant interacted even more efficiently with CIP2A; almost as efficiently as the wild type.

- *The TOPBP1 siRNA results in reduced CIP2A expression levels (Fig. S2D), and the authors therefore state in line 142 that it is not possible to determine on basis of the data presented in Fig. 3D whether loss of the TOPBP1-CIP2A and/or reduced CIP2A expression affects CIP2A recruitment to IR-foci. This could potentially be resolved by performing TOPBP1 RNAi in the +FLAG-CIP2A cells, which strongly overexpress CIP2A (Fig. 3A).*

We would like to thank Reviewer #2 for this elegant idea. We did as he/she suggested and in addition, we used a second siRNA against TOPBP1, targeting the 3' UTR (siTOPBP1 #2). The original siRNA (siTOPBP1 #1) unfortunately also efficiently downregulated the overexpressed Flag-CIP2A (new Fig. 3d). However, siTOPBP1 #2 did not. In fact, transfection with this siRNA for some reason led to an increased expression of Flag-CIP2A. Nevertheless, regardless of the effect on CIP2A expression, both siRNAs efficiently downregulated TOPBP1 and essentially abrogated CIP2A foci in mitosis. We therefore conclude that for efficient recruitment to sites of mitotic DSBs CIP2A and TOPBP1 are interdependent.

- *Related to the previous point, the TOPBP1 D813-892 mutant shows a mildly reduced interaction with CIP2A (Fig. 1D), and is not recruited to IR-foci (Fig. 1B). The lack of CIP2A localization to IR-foci, shown in Fig. 3E, could therefore be explained by either a requirement for TOPBP1 localization at the DSB-site, or a requirement for interaction of CIP2A with TOPBP1 (Fig. 3E). It is thus not possible to conclude, as the authors do in line 151, that CIP2A recruitment is dependent on its interaction with TOPBP1. This emphasizes again that the authors should map the interaction domain in more detail and repeat these experiments with a TOPBP1 mutant that fails to interact with CIP2A.*

As pointed out above, we have now included several additional deletion mutants in our analysis. All of them show at least partial interaction and recruitment defects. Only the wild type that contains both conserved amino acid patches optimally interacted with CIP2A and allowed efficient recruitment of CIP2A and TOPBP1 to IRIF on mitotic chromosomes. We therefore conclude that CIP2A and TOPBP1 recruitment is dependent on their interaction. The only piece of data arguing against this interpretation is the Δ 813-892 mutant that still (sub-optimally) interacted with CIP2A in our co-IP experiments, while being completely defective for recruitment. This does not quite add up. However, given that mutation of three amino acids in this region led to completely defective interaction in a yeast to hybrid system⁷, we hypothesize that the observed interaction of this mutant may be an over-expression artifact in our co-IP system. Under these conditions, the upstream interacting region (aa774-789) may be sufficient to mediate partial interaction in cell extracts *in vitro*. However, in cells, this residual interaction may not be sufficient to mediate recruitment. The observation that deletion of both conserved patches completely abrogates both interaction and recruitment in our opinion allows the conclusion that recruitment of CIP2A and TOPBP1 depends on their interaction.

- How can TOPBP1 foci be dependent on CIP2A (see Figure 4A), since TOPBP1 expression is not affected and TOPBP1 can still interact with MDC1 following CIP2A loss (see Supplementary figure 2D and Figure 4H, respectively)? The authors should clarify this experimentally to provide mechanistic insight into how CIP2A contributes to TOPBP1 foci formation.

This is a key issue, and we believe that we can provide some answers to this question in the revised version of our manuscript. Our data suggest that while interaction between MDC1 and TOPBP1 is essential, it is not sufficient for TOPBP1 foci formation. This was already clear from the data presented in our previous study⁹ where we showed that even though the interaction between MDC1 and TOPBP1 is constitutive and occurs throughout the cell cycle, it is insufficient to recruit TOPBP1 to sites of damage in interphase in the absence of RNF8-RNF168-53BP1 cascade. In the present manuscript we provide strong evidence that a ternary complex composed of MDC1, TOPBP1 and CIP2A needs to form to efficiently recruit TOPBP1 (and CIP2A) to sites DSBs in mitosis. We previously showed that interaction between MDC1 and TOPBP1 is not stimulated by DNA damage⁹. In this paper we show both *in vitro* and in cells that interaction between TOPBP1 and CIP2A is also not stimulated by DNA damage (see Fig. 1e and Fig. 2f). However, interaction between CIP2A and MDC1 is stimulated by DNA damage, both *in vitro* and in cells (see Fig. 5c,d). Thus, TOPBP1, despite being able to interact both with MDC1 and CIP2A in a constitutive manner, does not appear to simply bridge the two proteins to form a stable ternary complex. Instead, formation of a ternary complex occurs via a process that is stimulated by DNA damage. We currently do not understand how this works on a mechanistic level and several possibilities exist. For example, it is possible that CIP2A and MDC1 may interact directly in a DNA damage-dependent manner. Another possibility is that CIP2A or TOPBP1 (or both) may undergo conformational changes in response to DNA damage, which may be required for the assembly of an efficient ternary complex with MDC1 and thus, for recruitment to sites of DNA damage. Finally (and this is currently our favorite hypothesis because we do have preliminary evidence supporting it), it is possible that CIP2A and TOPBP1 form large multimeric assemblies (sometimes termed 'bio-molecular condensates') only in the presence of DNA damage. Such DNA damage-stimulated self-assembly may be required for efficient interaction with MDC1 and thus, for recruitment to sites of DNA damage. It appears that under physiological conditions, CIP2A and TOPBP1 accumulation is strictly dependent on DNA damage. However, we and others recently demonstrated that when TOPBP1 is overexpressed or forced to self-associate, accumulation can occur even in the absence of DNA damage¹⁰⁻¹². Here, we show that under such conditions, CIP2A resists CRM1-mediated nuclear export and is partially retained in the nucleus of interphase cells, where it co-localizes with TOPBP1 condensates (Supplementary Fig. 6g). We added a section in the Discussion part of the revised manuscript where we address this issue.

- Based on their findings in Figure 5, the authors claim that CIP2A is a direct CRM1 ligand.

Although the available mass spectrometry interaction data may support this, the authors should verify if CRM1 and CIP2A interact in a manner dependent on its NES at 598-612.

We performed this experiment, and the data are presented in new Fig. 6e. Deletion of the NES at 598-612 in CIP2A only mildly reduced its interaction with CRM1. This suggests the existence of one or several additional NES in CIP2A; an interpretation that is also supported by the observation that deletion of the NES at 598-612 led to a more moderate nuclear retention of CIP2A when compared to treatment of cells with CRM1 inhibitors, especially LeptomycinB (compare Fig. 6a,b with Fig. 6c,d).

- The authors conclude that CIP2A and TOPBP1 interact at sites of DSBs in mitosis (line 113). However, they formally only show CIP2A/TOPBP1 colocalization in foci on DNA. The authors should at least repeat the experiments shown in Fig. 2A, B and include a γ H2AX stain to show that the CIP2A/TOPBP1 foci colocalize with sites of damage. Furthermore, IR can induce many kinds of damage, not only DSBs. To formally conclude the colocalization occurs at DSBs, the authors should do this after inducing bona-fide DSBs using nuclease systems such as AsiSI, Cas9 or FokI.

We added microscopic data showing γ H2AX, CIP2A and TOPBP1 co-localization (new Supplementary Fig. 2c). To formally demonstrate that CIP2A and TOPBP1 occurs at *bona-fide* DSBs, we used the FokI DSB reporter system developed by Roger Greenberg's lab. This was technically challenging because FokI induction leads to the activation of the G2/M checkpoint. Moreover, the nuclease activity is not efficient on condensed chromosomes. However, by carefully timing FokI induction with cell cycle arrest by Nocodazole, we managed to image a few mitotic cells with a clear focal mCherry-FokI signal on the chromatin. One example is shown in new Supplementary Fig. 2d. In these cells, the DSB induction by FokI presumably occurred after transition from G2 into mitosis, but before the completion of chromosome condensation. In those cells, both CIP2A and TOPBP1 accumulated at and co-localized with, the mCherry-FokI focus, whereas in interphase cells, only TOPBP1 accumulated at and co-localized with, the mCherry-FokI focus (new Supplementary Fig. 2d). These data thus formally demonstrate that CIP2A and TOPBP1 are recruited to *bona-fide* DSBs in mitotic cells and add further support to the conclusion that CIP2A is not recruited to sites of DSBs in interphase cells.

- How CIP2A mechanistically contribute to DSB repair and genome stability maintenance remain rather unclear. To address this further the authors should show whether CIP2A contributes to TOPBP1-dependent bridging of MDC1-Bound DSBs Acquired during Mitosis as in their previous work, Pia Amata-Leimbacher et al., Mol. Cell (2019).

We previously showed that a subset of TOPBP1 filaments on mitotic chromosomes occur between two MDC1 foci. Other TOPBP1 structures co-localized with MDC1 and a third

category of TOPBP1 structures appeared as filaments outside of the chromatin mass⁹. In the absence of CIP2A, all the TOPBP1 structures in mitosis disappear (Fig. 3b,c). It is therefore impossible to assess whether bridging of MDC1 foci by TOPBP1 is specifically abrogated in the absence of CIP2A. As in the absence of CIP2A no filaments remain, neither on chromatin nor outside of the chromatin mass, we conclude that DSB bridging/tethering is dependent on CIP2A. However, we would like to point out that the DSB bridging/tethering model is still predominantly based on indirect evidence.

Minor Comments

- *"Inspection of one of these DNA damage chemogenetic datasets revealed a significant drug sensitivity correlation between MDC1 and CIP2A (Supplementary Fig. 1d)." How do the authors know this is a significant correlation? No numbers/statistics are provided to warrant this conclusion.*

We have now included the correlation coefficient (0.76) in the text and the Figure legend. The correlation cutoff in the Network analysis was set to 0.7. Similar drug sensitivity correlations with MDC1 included H2AX (0.79), MCPH1 (0.76), NHEJ1 (0.71) and RNF168 (0.71), all proteins known to physically or genetically interact with MDC1. This indicates drug sensitivity profiles upon loss of MDC1 and CIP2A do significantly correlate.

- *In Fig. 3B, to be able to fully appreciate the extent of colocalization, the authors should include pictures of the individual channels, like they do for the other IF-panels.*

We have now changed this image accordingly

- *Whereas all IF-pictures demonstrating colocalization between CIP2A and TOPBP1 are of cells in prophase, the siTOPBP1 example cell shown in Fig. 3D has clearly progressed to pro-metaphase and appears close to metaphase. For a fairer comparison of the siTOPBP1 condition to the other data, the authors should show a cell that is in prophase, like the other example cells shown.*

We replaced these panels with panels that show a cell in pro-metaphase, like the control panels.

- *A western blot associated to figure 3E, showing the expression of the TOPBP1 variants, is missing.*

This issue was also raised by Reviewer #1, A Western blot showing the expression of the TOPBP1 variants is now included (new Supplementary Fig. 4b)

- Also on Figure 3e, was the expression of CIP2A affected in cells expressing TOPBP1 Δ 813-192? If so, reduced CIP2A foci formation in these cells may also stem (partially) from this effect. This can be clarified by Western blot analysis (as in Supplementary figure 2d).

In the new TOPBP1 complementation analysis, we used siTOPBP #2 (the siRNA targeting the 3'-UTR of TOPBP1) to knock-down endogenous TOPBP1. This siRNA does not reduce CIP2A expression levels (see Fig. 3d). In addition, expression of the various TOPBP1 deletion mutants (including Δ 813-192) do not affect CIP2A expression (see Fig. 4a).

- In line 110, line 148 and line 149 the TOPBP1 D813-892 mutant is inadvertently written as D813-192 mutant.

This mistake has been corrected.

- Figures 1F and 4E, did secondary antibody controls only show no signals in the PLA assay?

We did the secondary antibody only controls and they did not show any signal in the PLA assay. However, based on critical comments by Reviewer #1, we now also included a control in which one of the interaction partners of CIP2A (TOPBP1 and MDC1, respectively), were depleted by siRNA (in the case of TOPBP1) or knocked-out by CRISPR/Cas9 (in the case of MDC1). We feel that this is the better control than the secondary antibody only control and thus, we decided to include it in the revised version of the manuscript (new Figs. 2f, 5d and 6f).

- A western blot is missing to validate (loss of) MDC1 expression in the DMDC1 cells with and without GFP-MDC1 WT or mutant (Fig. 4).

This control Western blot is now included (new Supplementary Fig. 5e)

- To clarify figure 7A, the figure legend should include what each symbol represents, in addition to explaining this in the materials and methods section.

We have now changed this graph completely. Instead of symbols, we now show columns and error bars that represent the proportion of cells that have MNi and the 95% confidence intervals of the proportions, respectively. We also plotted the -IR and +IR samples side by side in the same graph.

Reviewer #3 (Remarks to the Author):

Review comments

of "The CIP2A-TOPBP1 complex safeguards chromosomal stability during mitosis"

This manuscript by Mara De Marco Zompit et al. describes a study which identified CIP2A as a TOPBP1-MDC1 interacting protein that promotes TOPBP1 recruitment to sites of mitotic DSBs in mitosis. Cells lacking CIP2A display increased micronuclei formation, DSB repair defects, and chromosomal instability. CIP2A is actively exported from the cell nucleus in interphase but, upon nuclear envelope breakdown at the onset of mitosis, gains access to chromatin. Therefore, authors propose CIP2A-TOPBP1 as a mitosis-specific genome maintenance complex.

CIP2A is well-documented oncogene and a presumed therapeutic target for breast cancer. The protein is also a well-known PP2A inhibitor and was shown to stabilize MYC in human malignancies. The discovery of CIP2A's role in maintaining genome stability has implications on the use of CIP2A inhibitor in a clinical context and therefore is interesting and important. Mara De Marco Zompit et al. propose CIP2A as a TOPBP1-interacting protein that promotes TOPBP1 recruitment to sites of DSBs specifically in mitosis in MDC1-dependent manner. TOPBP1 is known to specifically interact with Casein kinase 2 (CK2)-phosphorylated MDC1 and this interaction is required for mitotic recruitment of TOPBP1 to DSB sites by MDC1. CIP2A is a well-established PP2A inhibitor, and the authors propose that CIP2A promotes TOPBP1 mitotic recruitment to DSB site by MDC1 without considering the possibility that CIP2A inhibits PP2A activity in dephosphorylating MDC1?

This manuscript by Mara De Marco Zompit et al. is carefully written, most experiments are well designed, and most data are clean. However, there is a consistent lack of proper controls and DNA damage/mitotic markers that negatively impacts the quality of the manuscript. Figures are not labeled clearly and are not presented in a way that is easy to understand. The lack of statistical (?) analysis on about half of all figures and the missing mechanistic connection between CIP2A-TOPBP1 interaction and increased micronuclei to cell death leave the story half finished. Before reaching their conclusion that CIP2A binds to TOPBP1-MDC1 complex to promote TOPBP1 recruitment to sites of mitotic DSBs to promote genome stability, there are some main issues that need to be addressed:

We thank Reviewer #3 for this overall evaluation and for his/her assessment that our study is interesting and important. Also, we are glad to hear that this reviewer thought that our experiments were well designed, and most data were clean. We also appreciate this reviewer's critical comments on the issue of whether CIP2A, which is an established PP2A inhibitor, could exercise its functions in the mitotic DDR by inhibiting PP2A. We have thoroughly addressed this issue in the revised version of our manuscript (see below). We also added missing controls, especially regarding DNA damage and mitotic markers. We expect that the manuscript thus gained in clarity and the Figures are now more clearly presented and easier to understand.

Major concern:

1. Figure 1. To demonstrate that CIP2A directly interacts with TOPBP1, endogenous IP and in

in vitro binding experiments with recombinant protein are needed. Also, the authors need to demonstrate whether CIP2A's ability to inhibit PP2A is required for the interaction between CIP2A and TOPBP1 and between TOPBP1 and MDC1.

We added *in vitro* binding experiments using recombinant proteins to the revised version of our manuscript (new Fig. 1d). The purified region of TOPBP1 that pulled-down CIP2A from cell extracts (amino acids 740-899) also pulled down the bacterially expressed purified ArmRD of CIP2A. We could not test binding to the C-terminal coiled-coil region of CIP2A because it is insoluble when expressed in isolation, both in bacteria and in mammalian cells. We also added several experiments to test if CIP2A acts as PP2A inhibitor in the mitotic response to DNA damage. An attractive hypothesis is that CIP2A could mediate the interaction between MDC1 and TOPBP1 (which is dependent on phosphorylation of MDC1), by antagonizing MDC1 de-phosphorylation by PP2A. However, MDC1 phosphorylation on Ser168 and on Ser196 (the two TOPBP1 interaction sites) is neither affected by deletion of CIP2A nor by treatment of cells with the small molecule PP2A inhibitor LB-100 (Fig. 5g). We also show directly that CIP2A is not required for the interaction between MDC1 and TOPBP1 (Fig. 5h). Finally, we tested if inhibition of PP2A by LB-100 could rescue TOPBP1 foci in the absence of CIP2A. This was based on the idea that if CIP2A promoted TOPBP1 recruitment to sites of mitotic DSBs by inhibiting PP2A, replacing CIP2A with small molecule inhibitors against PP2A would have the same effect and rescue TOPBP1 recruitment. However, this was not the case (Supplementary Fig. 5e).

2. Does the TOPBP1-CIP2A interaction increase after IR and over mitosis in IP-WB?

This issue was also raised by Reviewer #2 and is now addressed in new Fig. 1e)

3. Figure 2. As shown in figure 2A, TOPBP1 and CIP2A also interact during mitosis in the absence of IR although in less foci. Authors need to determine if the TOPBP1-CIP2A interaction during mitosis is co-localized with gamma H2Ax foci, or centrosome and unattached kinetochores. Increased DNA damage can increase the number of unattached kinetochores or can delay kinetochore attachment to mitotic spindle, which can lead to a similar increase in mitotic foci formation.

We have added these control experiments in the revised version of the manuscript. They show that CIP2A foci co-localize with the DSB marker γ H2AX (new Supplementary Fig. 1c). They also show that CIP2A foci do not co-localize with centromeric markers. Shown here is ACA, but we also tested the marker CENPA with similar results (data not shown). CIP2A does however co-localize with centrosomes (as does TOPBP1; Fig. 2a and Supplementary Fig. 2a), which has been observed before¹³. We also added data showing that CIP2A and TOPBP1 are recruited to *bona fide* DSBs in mitosis (new Supplementary Fig. 2d).

4. Figure 3. Please demonstrate that loss of CIP2A or TOPBP1 does not lead to a corresponding change of TOPBP1 or CIP2A total protein level.

These data were already included in the first version of the manuscript (old Supplementary Fig. 2d). They showed that loss of CIP2A did not affect TOPBP1 protein level. However, loss of TOPBP1 by siRNA led to a reduction of CIP2A protein levels. To overcome this obstacle, we used our complemented RPE-1 CIP2A knock-out cell line that overexpresses recombinant Flag-tagged CIP2A (see also response to Reviewer #2). In addition, we used a second siRNA against TOPBP1 in the revised version of the manuscript. This second siRNA targets the 3'-UTR of TOPBP1. Surprisingly, this siRNA had the opposite effect on CIP2A protein levels, as it increased them (Fig. 3d). We currently do not understand these opposite effects of the TOPBP1 siRNAs on CIP2A protein levels. However, since both siRNAs completely abrogate CIP2A foci in mitosis, and since re-expression of recombinant full-length TOPBP1 rescues CIP2A foci, we conclude that TOPBP1 is required for CIP2A recruitment, and does not simply downregulate CIP2A protein levels to such an extent that it cannot be detected by immunofluorescence anymore.

5. Figure 3d. Please demonstrate mitotic DSB foci number is not affected by loss of CIP2A or TOPBP1 with gamma-H2Ax foci.

We performed this control experiments and the data can be found in new Supplementary Fig. 3f. They show that γ H2AX foci in mitosis are not affected by CIP2A loss. However, we found that γ H2AX foci in mitosis are mildly increased by loss of TOPBP1. We think that the increase in γ H2AX in the absence of TOPBP1 is likely reflecting its roles in DNA replication and ATR activation. It is well established that increased replication stress can lead to DNA breaks in mitosis.

6. Figure 4c. MW of GFP-MDC1 pulled down is higher than input.

We replaced this pull down with a new experiment (new Fig. 5c)

7. Figure 4f. Does MDC1S168E/S196E mutant bring TOPBP1 to mitotic foci? Will CIP2A loss still impact TOPBP1 mitotic foci formation in the presence of MDC1S168E/S196E?

We previously showed that TOPBP1 binds to MDC1 phosphorylated on Ser168 and Ser196, via its N-terminal BRCT domains (BRCT1 and 2)⁹. It is well established that phospho-Ser mimicking Ser to Glu mutation does not rescue BRCT binding because X-ray structural analysis clearly revealed that BRCT domains require a phosphate group as ligand and are not binding to glutamic acids¹⁴. As a consequence, we did not perform this experiment as in would have been futile.

8. *Figure 4h. Please include IP-WB in mitotic cells as Figure 5 indicates CIP2A does not colocalize with TOPBP1 during interphase.*

While we understand why the Reviewer made this comment, one of the main points of this manuscript is that CIP2A-TOPBP1 interaction is controlled during the cell cycle by nuclear export of CIP2A. Extract preparation for IP-WB destroys the nuclear envelope, thus abrogating spatial separation of TOPBP1 and CIP2A. We show that CIP2A and TOPBP1 do co-localize in interphase, when CIP2A is forced in the nucleus (Supplementary Fig. 6g,h), which strongly suggests that their interaction is not directly controlled by the cell cycle, but indirectly, through nuclear export of CIP2A in interphase. As a consequence, the suggested experiment is unlikely to be informative and therefore we opted not to perform it.

9. *Figure 5, 6. If 3XNLS tag is added to CIP2A-DeltaNES mutant, will that bring CIP2A to nucleus and regulate MDC1-TOPBP1 interaction after IR induced DNA damage?*

We do not entirely understand the purpose of this suggested experiment. We show that treatment of cells with CRM1 inhibitors brings CIP2A in the nucleus in interphase, where it co-localizes with TOPBP1 (Supplementary Fig. 6g). We also see this when we express CIP2A Δ NES in CIP2A knock-out cells, albeit much weaker (data not shown). We previously showed that MDC1 controls TOPBP1 recruitment by direct interaction exclusively in mitosis⁹. In interphase cells, TOPBP1 recruitment is controlled by the RNF8-RNF168-53BP1 cascade and direct interaction with 53BP1 or, in the case of rDNA breaks, by Treacle^{14,15 11}. Finally, we show that CIP2A does not regulate MDC1-TOPBP1 interaction, not even in mitosis (Fig. 5h).

10. *Figure 6. Can you show interphase TOPBP1 foci with 1Gray IR?*

While we do not entirely understand the purpose of this question, we assume that the Reviewer meant that 1 Gy of IR may not be a sufficiently high dose to induce TOPBP1 foci in interphase cells. This is however not the case. 1 Gy is more than enough to induce TOPBP1 foci in interphase cells. The reason why we used 3 Gy in interphase cells instead of 1 Gy is that we opted for doses with comparable toxicity in both interphase and mitosis (see also below).

11. *Figure 7a. Please indicate how many cells were quantified. Also plot -IR and +IR in the same graph and perform statistical analysis over CIP2A-/- cells before and after IR.*

We have revised this analysis and changed the graph type. Following this reviewer's suggestion, we now show -IR and +IR results on the same graph. We plotted the proportion (columns) and 95% confidence intervals of the proportion (error bars) of cells with MNi. This shows that while there is a statistically significant difference in the proportion of cells with MNi between the cell lines (95% confidence intervals do not overlap), the slight increase in the

proportion of cells with MNi after IR treatment is not statistically significant (95% confidence intervals overlap). We therefore revised the interpretation of this experiment also in the text. We also indicated how many cells were quantified per condition in the Figure legend.

12. Figure 7e, IR dose changed to 0.5 Gray again? Why? Please keep the IR dose consistent or show results in dosage and timepoint.

In this experiment, cells arrested in mitosis were irradiated. Mitotic cells are extremely radio-sensitive. Even treatment with 1 Gy leads to significant cell death within a few hours. We therefore used 0.5 Gy in this experiment to keep toxicity at a tolerable level. Since all the groups in this experiment were treated with the same dose of IR (0.5 Gy), and since we do not compare these groups to other groups that were treated a different dose of IR, we do not see a problem here.

Minor issue:

1. Figure 1a. Please indicate BRCT domain on schematic for assigned number 0-8, also include aa sequence number on schematic.

We have changed the Figure accordingly

2. Figure 1b. There is a lack of quantification and statistical analysis. Also, the authors need include a mitotic marker such as p-Histone H3 and a DNA damage marker such as gamma H2Ax.

In this experiment we tested the effects of various deletions on TOPBP1 foci formation in interphase cells and mitotic cells. For the interphase cell analysis, we used unsynchronized cell populations, for mitotic cell analysis, we arrested cells in mitosis by Nocodazole. We identified mitotic cells based on the DAPI channel, which reveals condensed chromosomes as a clear marker for mitotic cells. We do not believe that p-Histone H3 staining is necessary to mark mitotic cells in this experiment. To make this point clearer, we included a control experiment with the p-Histone H3 marker. This experiment clearly shows how well mitotic cells can be distinguished from interphase cells based on DAPI staining (new Supplementary Fig. 2b). Moreover, we also include control experiments that show that the TOPBP1 foci co-localize with DSB marker γ H2AX (Supplementary Fig. 2c). As the experiment in Fig. 1b was designed to screen for sequence motifs required for TOPBP1 foci formation in mitosis, we do not think that detailed quantification and statistical analysis is required at this point. This analysis was done in later experiments where we show (among other things) that deletion of aa 751-899 generates a statistically significant defect in TOPBP1 accumulation at sites of mitotic DSBs (see Fig. 4c).

3. Figure 1b. The Mitosis and Interphase [what] use different doses of IR? Please use same dose to exclude the possibility that the difference is due to the difference in IR dose.

We decided to use 3 Gray of IR in unsynchronized cell populations where most cells are in interphase, to match 1 Gy dose in cells arrested in mitosis (see also above). We think results are easier to interpret if cells are treated with doses that induce comparable genotoxicity.

4. Figure 2a, 2b. The Mitosis and Interphase [what] use different doses of IR? Please use same dose to exclude the possibility that the difference is due to the difference in IR dose.

We did not use different doses of IR in this experiment. The left graph shows untreated cells, the right graph shows cells treated with 1 Gy of IR. This is clearly stated in the Figure legend.

5. Figure 2. There is a lack of quantification and statical analysis. Also, the authors need include a mitotic marker such as gamma tubulin or mad2 and a DNA damage marker such as gamma H2Ax.

We have added additional quantification and statistical analysis to this image. The take-home message of this image is that CIP2A and TOPBP1 interact and co-localize at sites of DSBs exclusively in mitosis. This is now quantified by Squassh (Fig. 2d) and PLA (Fig. 2f). We also included controls for DSB and mitotic markers in Supplementary Fig. 2.

6. All figure with images, label the scale bar on image.

We labelled the scale bars in the Figure legends and not on the images, as labeling them on the images is potentially covering important data. This is also in agreement with the journal policies for Figure preparation.

7. Figure 2D. there is a lack of a mitotic marker like p-Histone H3.

Former Fig. 2d is a quantification graph of the Squassh experiment for CIP2A and TOPBP1 co-localization in interphase and mitosis. In this experiment, we selected mitotic cells based on the DAPI channel, which reveals condensed chromosomes as a clear marker for mitotic cells. We do not believe that p-Histone H3 staining is necessary to mark mitotic cells in this experiment. To make this point clearer, we included a control experiment with the p-Histone H3 marker. This experiment clearly shows how well mitotic cells can be distinguished from interphase cells based on DAPI staining (new Supplementary Fig. 2b)

8. Figure 3B. Include MAD2 and r-tubulin staining as localization control for centromere and centrosome (mitotic spindle).

We have now included DSB markers (γ H2AX), centromere and centrosome markers that yield a good picture as to where CIP2A is located during mitosis (see new Supplementary Fig. 2). We have a maximum of four channels available on our confocal microscope setup, so MAD2

and r-tubulin staining along with DAPI, CIP2A and TOPBP1, as suggested here, is technically impossible.

9. *CIP2A and TOPBP1 are both required to recruit one another to mitotic foci, the title in result section did not reflect the fact that TOPBP1 is also required for CIP2A recruitment to mitotic foci and therefore is misleading.*

We have changed the title of this result section to: 'TOPBP1 accumulation at sites of mitotic DSBs is dependent on CIP2A and *vice versa*', which is now reflecting our observations more accurately.

10. *Figure 6b, statistical analysis needed.*

Statistical analysis was added

11. *Figure 7c, please clearly label on the figure what the red dots indicating.*

This information was added to the Figure.

12. *Figure 7d, statistical analysis needed.*

Statistical analysis was added

13. *Figure 7e, please label clearly on figure what the red dots indicating or use red font to indicate to be consistent with green gammaH2Ax and blue DAPI.*

We have exchanged the colors in the image segmentation panels of this Figure to that they match the colors in the micrographs (blue = DAPI, green = γ H2AX). It should now be clearer what they indicate.

References

1. Blackford, A. N. & Stucki, M. How Cells Respond to DNA Breaks in Mitosis. *Trends in biochemical sciences* 1-11 (2020).
2. Deng, L. et al. Mitotic CDK Promotes Replisome Disassembly, Fork Breakage, and Complex DNA Rearrangements. *Mol Cell* **73**, 915-929.e6 (2019).
3. Heijink, A. M. et al. Sister chromatid exchanges induced by perturbed replication are formed independently of homologous recombination factors. (2021).

4. Royou, A., Gagou, M. E., Karess, R. & Sullivan, W. BubR1- and Polo-Coated DNA Tethers Facilitate Poleward Segregation of Acentric Chromatids. *Cell* **140**, 235-245 (2010).
5. Adam, S. et al. CIP2A is a prime synthetic-lethal target for BRCA-mutated cancers. *Nat Cancer* **in press**, (2021).
6. Pedersen, R. T., Kruse, T., Nilsson, J., Oestergaard, V. H. & Lisby, M. TopBP1 is required at mitosis to reduce transmission of DNA damage to G1 daughter cells. *210*, 565-582 (2015).
7. Adam, S. et al. The CIP2A–TOPBP1 axis safeguards chromosome stability and is a synthetic lethal target for BRCA-mutated cancer. *Nature Cancer* (2021).
8. Laine, A. et al. CIP2A Interacts with TopBP1 and Drives Basal-Like Breast Cancer Tumorigenesis. *Cancer Research* **81**, 4319-4331 (2021).
9. Leimbacher, P.-A. et al. MDC1 Interacts with TOPBP1 to Maintain Chromosomal Stability during Mitosis. *Molecular Cell* **74**, 571-583.e8 (2019).
10. Sokka, M., Rilla, K., Miinalainen, I., Pospiech, H. & Syvaaja, J. E. High levels of TopBP1 induce ATR-dependent shut-down of rRNA transcription and nucleolar segregation. *Nucleic acids research* **43**, 4975-4989 (2015).
11. Mooser, C. et al. Treacle controls the nucleolar response to rDNA breaks via TOPBP1 recruitment and ATR activation. *Nature Communications* **11**, 1-16 (2019).
12. Frattini, C. et al. TopBP1 assembles nuclear condensates to switch on ATR signaling. *Mol Cell* (2021).
13. Kim, J. S., Kim, E. J., Oh, J. S., Park, I. C. & Hwang, S. G. CIP2A modulates cell-cycle progression in human cancer cells by regulating the stability and activity of Plk1. *Cancer Res* **73**, 6667-6678 (2013).
14. Bigot, N. et al. Phosphorylation-mediated interactions with TOPBP1 couple 53BP1 and 9-1-1 to control the G1 DNA damage checkpoint. *eLife* **8**, 3894 (2019).
15. Cescutti, R., Negrini, S., Kohzaki, M. & Halazonetis, T. D. TopBP1 functions with 53BP1 in the G1 DNA damage checkpoint. *The EMBO Journal* **29**, 3723-3732 (2010).

REVIEWER COMMENTS

Reviewer #1 (Remarks to the Author):

The manuscript has been significantly improved by the author's revisions. However, I have some minor concerns specifically relating to the validity of carrying out statistical analysis on $n=2$ independent experiments. In my view, it does not matter how many data points are collected per experiment, a minimum of $n=3$ independent experiments must be carried out to be able to calculate a mean, add error bars and carry out statistical analysis. Furthermore, several Western blots look like they have been duplicated. If it is the case that a blot has been probed with one antibody then stripped and re-probed with another then it should be stated as such.

Minor comments:

Figure 2f: The number of independent experiments needs to be stated.

Supp Figure 3f: How can error bars be added and stats carried out on $n=2$ independent experiments?

Figure 5c. The MDC1 and HA input Westerns for the WT and S168/196A MDC1 look like the same blot with the HA just being a slightly darker exposure.

Figure 5d. How can stats carried out on $n=2$ independent experiments?

Figure 6b & 6d: How can error bars be added and stats carried out on $n=2$ independent experiments?

Figure 6e. The Flag and CIP2A blots for the input and Flag-IP look like the same blots but with being a slightly darker exposure of the other.

Figure 6g and 6i: The number of independent experiments needs to be stated.

Supp Figure 6e: The number of independent experiments needs to be stated.

Figure 7b, 7d and 7f: The number of independent experiments needs to be stated.

Reviewer #2 (Remarks to the Author):

The authors have carefully considered my comments, performed several experiments to improve the manuscript, and thoroughly revised the manuscript. Nevertheless, there are a few remaining issues that I think need to be addressed before publication of the manuscript can be considered for publication.

- Based on new co-IP experiments the authors conclude that the interaction between TOPB1 and CIP2A is increased after nocodazole treatment. However, I doubt whether this conclusion can be drawn from the current data as it seems as if the untreated control cells and nocodazole-treated cells were from different experiments and/or loaded on a different blot. If so, these cannot be compared, hampering the conclusion that the TOPB1-CIP2A interaction is increased after nocodazole. Ideally, one blot showing both untreated control cells and nocodazole-treated cells is presented, accompanied by a quantification from at least 3 independent experiments.

- How can the TOPB1-CIP2A interaction be observed in cycling, untreated cells which are mostly in cell cycle phases other than M phase (only a few percent of the cells will be in M phase), and in IR-treated cells which become arrested mainly in G2 phase? I am asking because outside M phase CIP2A has been shown to localize to the cytoplasm and not the nucleus to which TOPB1 localizes.

- Related to the previous point, the TOPBP1-CIP2A interaction occurs in M phase at centrosomes but is not enhanced by IR. How specific/relevant is than the interaction for DNA damage in mitosis?

- The authors state in line 173-176 "CIP2A efficiently accumulated in foci only in the presence of wild type TOPBP1, but not in the presence of deletion mutants, thus indicating that the efficient recruitment of both of these proteins to sites of mitotic DNA breaks is dependent on their interaction (Fig. 4b,c, Supplementary Fig. 4b)." However, the domain mapping and recruitment studies suggest that TOPBP1 774-798delta doesn't interact with but recruits CIP2A to DNA damage, whereas TOPBP1 813-892delta interacts with but doesn't recruit CIP2A to DNA damage. (Figure 4 and reviewer only figure in rebuttal). This suggest to me that interaction and recruitment are not interdependent, contrary to the author's conclusion in line 173-176.

- The reciprocal co-IPs for TOPBP1 813-892delta seem inconsistent as Figure 4a suggests this mutant is partially impaired in the interaction with CIP2A, while the reviewer only figure in the rebuttal suggest the interaction is largely intact. These are important data suggesting that TOPBP1 813-892delta may not be completely interaction-defective. These results should be presented in the supplementary information rather than the rebuttal.

- Labeling at the X-axis in Figure 4C doesn't match that in Figure 4B (lables above the images).

- The legend of figure 4C legend states "Quantification of GFP-TOPBP1/CIP2A foci per cell", but is it GFP-TOPBP1 or CIP2A that was quantified?

Reviewer #3 (Remarks to the Author):

The author successfully addressed some of the major concerns and most of the minor concerns. The especially important one is to exclude the alternative hypothesis that CIP2A's ability to inhibit PP2A might be required for the interaction between CIP2A and TOPBP1 and between TOPBP1 and MDC1.

However, Author refuses to address the concern that the study is using an inconsistent dose of IR between interphase cells and mitotic cells. Different doses of IR lead to different biological consequences in vivo and therefore comparing different doses of IR between interphase and mitosis cells can be misleading. I do understand that mitotic cells are more sensitive to IR due to lack/altered DNA damage repair capacity, however, the author can choose to use the same low dose IR in interphase cells be make the biological outcome more comparable.

Response to Reviewers

1. Overview

We thank the reviewers for evaluating our revised manuscript. Below, we provide a point-by-point response to all specific questions and concerns. Reviewers' comments are italicized.

2. Point-by-point response

Reviewer #1 (Remarks to the Author):

The manuscript has been significantly improved by the author's revisions. However, I have some minor concerns specifically relating to the validity of carrying out statistical analysis on $n=2$ independent experiments. In my view, it does not matter how many data points are collected per experiment, a minimum of $n=3$ independent experiments must be carried out to be able to calculate a mean, add error bars and carry out statistical analysis. Furthermore, several Western blots look like they have been duplicated. If it is the case that a blot has been probed with one antibody then stripped and re-probed with another then it should be stated as such.

We are glad to read that this reviewer thinks our manuscript has significantly improved. As for the comments on our statistical analysis, we don't understand the concerns. We clearly stated in the Figure legends what bars and error bars represented, and we described how statistical analysis was performed in the Material & Methods section. We also clearly indicated the sample size for each experiment, which in all cases, was much more than $n=2$. We calculated the mean and SD from all the data points shown in the graphs, and we pooled those data points from 2-3 independent experiments to take the experimentation variation into account. The data for all the graphs in the manuscript is provided in a Source Data file. It can be used by this reviewer to reproduce the statistical analysis with GraphPad Prism software (or any other statistical analysis software).

Second, we don't understand what this reviewer means by "several Western blots look like they have been duplicated". We have not duplicated any data. It is common practice to strip and re-probe Western blots with different antibodies and we are not aware that this needs to be stated every time (there is no mentioning in Journal's guidelines about this). The full-size scans of the Western blots can be found in the Source Data file. From these full-size scans (particularly the unspecific low molecular weight bands), it will become clear that we did not duplicate any data

Minor comments:

Figure 2f: The number of independent experiments needs to be stated.

This information has been added to the Figure legend.

Supp Figure 3f: How can error bars be added and stats carried out on n=2 independent experiments?

The sample size is clearly indicated in this experiment. It is n=61 for siCtrl, n=52 for siCIP2A, n=62 for siCtrl and n=68 for siTOPBP1. Mean and SD were calculated from all those data points, which have been pooled from two independent experiments. The source data for this graph is provided in the Source Data file.

Figure 5c. The MDC1 and HA input Westerns for the WT and S168/196A MDC1 look like the same blot with the HA just being a slightly darker exposure.

This is not the case. The primary data for these Western blots is provided in the Source Data file. On the full-size images of the exposed membranes, it is clearly visible that these are different blots (the HA blot shows much more unspecific low molecular weight bands). But since both antibodies recognize the same protein (HA-tagged MDC1), it is clear (and expected), that the Western blots look similar.

Figure 5d. How can stats carried out on n=2 independent experiments?

This is the same issue as already explained above for Supplementary Figure 3f. Again, statistical parameters and statistical tests were performed on all the data points. The source data for this graph is provided in the Source Data file.

Figure 6b & 6d: How can error bars be added and stats carried out on n=2 independent experiments?

This is the same issue as already explained above for Supplementary Figure 3f. Again, statistical parameters and statistical tests were performed on all the data points. The source data for this graph is provided in the Source Data file.

Figure 6e. The Flag and CIP2A blots for the input and Flag-IP look like the same blots but with being a slightly darker exposure of the other.

This is the same issue as explained for Figure 5c. The primary data for these Western blots is provided in the Source Data file. On the full-size images of the exposed membranes, it is clearly visible that these are different blots (the Flag and CIP2A Western blots show different low molecular weight unspecific bands). In addition, the difference also becomes apparent because in the Input Western blot that was probed with anti CIP2A antibody, a weak band appears also in the Mock lane (which corresponds to endogenous CIP2A). This band is not present in the Western blot probed with anti-Flag antibody, clearly showing that these membranes were probed with different antibodies and not with the same antibody at different exposure time.

Figure 6g and 6i: The number of independent experiments needs to be stated.

This information has been added to the Figure legend.

Supp Figure 6e: The number of independent experiments needs to be stated.

This information has been added to the Figure legend.

Figure 7b, 7d and 7f: The number of independent experiments needs to be stated.

This information has been added to the Figure legend.

Reviewer #2 (Remarks to the Author):

The authors have carefully considered my comments, performed several experiments to improve the manuscript, and thoroughly revised the manuscript. Nevertheless, there are a few remaining issues that I think need to be addressed before publication of the manuscript can be considered for publication.

- Based on new co-IP experiments the authors conclude that the interaction between TOPBP1 and CIP2A is increased after nocodazole treatment. However, I doubt whether this conclusion can be drawn from the current data as it seems as if the untreated control cells and nocodazole-treated cells were from different experiments and/or loaded on a different blot. If so, these cannot be compared, hampering the conclusion that the TOPBP1-CIP2A interaction is increased after nocodazole. Ideally, one blot showing both untreated control cells and nocodazole-treated cells is presented, accompanied by a quantification from at least 3 independent experiments.

We agree that this is a valid concern. We have now repeated this experiment four times and loaded all the different treatments on the same SDS gels so that all the treatments are processed on the same Western blot. We replaced the previous Fig. 1e with one representative example of the newly performed experiments. We then quantified the results by measuring the Flag and TOPBP1 band intensities in the Flag-IP blots with our BioRad ChemiDoc system. We calculated the ratio between Flag and TOPBP1 band intensities for each experiment. The data showed quite a bit of variation. In two experiments, Nocodazole treatment led to increased Flag/TOPBP1 band intensity ratios, in the other two experiments, this treatment led to decreased Flag/TOPBP1 band intensity ratios. We then combined all the quantifications in one graph (new Fig. 1f). The data show that there is not much of a difference in the Flag/TOPBP1 band intensity ratios both in the presence or absence of IR treatment and in the presence and absence of Nocodazole. We therefore conclude that neither DNA damage nor synchronization of cells in mitosis leads to a significant increase or decrease in the CIP2A-TOPBP1 interaction. This is consistent with one of the key take home messages of this paper: namely, that cell cycle regulation of

CIP2A-TOPBP1 interaction does not occur at the level of protein-protein interaction, but via spatial separation of the two proteins in interphase through CRM1-mediated nuclear export of CIP2A. We also changed the text in the manuscript according to these new data.

- How can the TOPBP1-CIP2A interaction be observed in cycling, untreated cells which are mostly in cell cycle phases other than M phase (only a few percent of the cells will be in M phase), and in IR-treated cells which become arrested mainly in G2 phase? I am asking because outside M phase CIP2A has been shown to localize to the cytoplasm and not the nucleus to which TOPBP1 localizes.

One of the key messages of our paper is that the CIP2A-TOPBP1 interaction is controlled during the cell cycle by nuclear export of CIP2A in interphase. Since in interphase, TOPBP1 is a nuclear protein and CIP2A is mostly localized in the cytoplasm, they do not interact, because they are spatially separated by the nuclear envelope. However, upon cell extract preparation, the nuclear envelope is destroyed and CIP2A and TOPBP1 are free to interact in the cell extract. That's why we detect the interaction by co-immunoprecipitation from cell extracts of unsynchronized cell populations, even though we don't see any interaction in fixed cells with intact nuclear envelope by the PLA assay (Fig. 6g and Supplementary Fig. 6c)

- Related to the previous point, the TOPBP1-CIP2A interaction occurs in M phase at centrosomes but is not enhanced by IR. How specific/relevant is than the interaction for DNA damage in mitosis?

We never claimed that TOPBP1 and CIP2A interact in M phase at centrosomes. We do occasionally see a punctuate PLA signal in untreated mitotic cells, but whether this signal originates from centrosomes is currently not clear, as we could so far not combine PLA staining with immunofluorescence. It is clear though that TOPBP1 and CIP2A both localize to centrosomes in mitosis. Yet, in contrast to DNA damage sites, their localization to centrosomes does not depend on the other protein (i.e. TOPBP1 localization to centrosomes does not depend on CIP2A and vice versa). Thus, there is the possibility that CIP2A and TOPBP1 do not interact at centrosomes but happen to localize there independently of each other and apart from each other. Further work is needed to clarify this issue, but we do not believe it is relevant for the current story, which does not focus on centrosomes but on DNA damage sites.

- The authors state in line 173-176 "CIP2A efficiently accumulated in foci only in the presence of wild type TOPBP1, but not in the presence of deletion mutants, thus indicating that the efficient recruitment of both of these proteins to sites of mitotic DNA breaks is dependent on their interaction (Fig. 4b,c, Supplementary Fig. 4b)." However, the domain mapping and recruitment studies suggest that TOPBP1 774-798delta doesn't interact with but recruits CIP2A to DNA damage, whereas TOPBP1 813-892delta interacts with but doesn't recruit CIP2A to DNA damage. (Figure 4 and reviewer only figure in rebuttal). This suggest to me that interaction and recruitment are not interdependent, contrary to the author's conclusion in line 173-176.

We are currently unable to fully explain these observations. However, it needs to be appreciated that the interaction data have been gathered by co-immunoprecipitation of overexpressed proteins from cell extracts. This is not the same as the localization data that were collected from intact fixed cells by fluorescence microscopy. It becomes more and more clear that DNA damage foci are controlled both by stoichiometric one-to-one binding (lock and key or induced fit) associations and by highly dynamic multivalent interactions (recently reviewed by Spegg and Altmeyer, 2021, DNA Repair 106). The first one would also be observable by co-immunoprecipitation, while the second one would not, because those highly dynamic multivalent interactions may be destroyed during cell extract preparation. It is thus possible that the region between 774-798 is mainly involved in relatively “rigid” one-to-one binding to CIP2A, while the regions between 813-891 is mostly involved in more dynamic multivalent interactions and/or TOPBP1 and CIP2A multimerization that manifests in cytologically discernible accumulation of the proteins in DNA damage foci. Both contribute to foci formation, because the 774-798 deletion mutant is clearly compromised, albeit not completely defective for accumulation at sites of DSBs.

- The reciprocal co-IPs for TOP1 813-892delta seem inconsistent as Figure 4a suggests this mutant is partially impaired in the interaction with CIP2A, while the reviewer only figure in the rebuttal suggest the interaction is largely intact. These are important data suggesting that TOPBP1 813-892delta may not be completely interaction-defective. These results should be presented in the supplementary information rather than the rebuttal.

We have included these data in the Supplementary Figures (new Supplementary Fig 4b).

- Labeling at the X-axis in Figure 4C doesn't match that in Figure 4B (lables above the images).

This has now been corrected.

- The legend of figure 4C legend states “Quantification of GFP-TOPBP1/CIP2A foci per cell”, but is it GFP-TOPBP1 or CIP2A that was quantified?

The foci were counted manually from the merge channel. In that regard, the statement in the figure legend is correct.

Reviewer #3 (Remarks to the Author):

The author successfully addressed some of the major concerns and most of the minor concerns. The especially important one is to exclude the alternative hypothesis that CIP2A's ability to inhibit PP2A might be required for the interaction between CIP2A and TOPBP1 and between TOPBP1 and MDC1.

However, Author refuses to address the concern that the study is using an inconsistent dose of IR between interphase cells and mitotic cells. Different doses of IR lead to different

biological consequences in vivo and therefore comparing different doses of IR between interphase and mitosis cells can be misleading. I do understand that mitotic cells are more sensitive to IR due to lack/altered DNA damage repair capacity, however, the author can choose to use the same low dose IR in interphase cells to make the biological outcome more comparable.

We explained in detail why we used different doses in interphase and mitotic cells in the previous rebuttal letter and since we never compared interphase cells to mitotic cells within the same experiment, we considered this a reasonable and acceptable choice.

REVIEWERS' COMMENTS

Reviewer #1 (Remarks to the Author):

No comments for the authors.

Reviewer #2 (Remarks to the Author):

The authors have carefully considered my comments, performed several experiments to improve the manuscript, and thoroughly revised the manuscript. Nevertheless, there are a few remaining issues that I think need to be addressed before publication of the manuscript can be considered for publication.

- How can the TOPBP1-CIP2A interaction be observed in cycling, untreated cells which are mostly in cell cycle phases other than M phase (only a few percent of the cells will be in M phase), and in IR-treated cells which become arrested mainly in G2 phase? I am asking because outside M phase CIP2A has been shown to localize to the cytoplasm and not the nucleus to which TOPBP1 localizes.

Reply author:

One of the key messages of our paper is that the CIP2A-TOPBP1 interaction is controlled during the cell cycle by nuclear export of CIP2A in interphase. Since in interphase, TOPBP1 is a nuclear protein and CIP2A is mostly localized in the cytoplasm, they do not interact, because they are spatially separated by the nuclear envelope. However, upon cell extract preparation, the nuclear envelope is destroyed and CIP2A and TOPBP1 are free to interact in the cell extract. That's why we detect the interaction by co-immunoprecipitation from cell extracts of unsynchronized cell populations, even though we don't see any interaction in fixed cells with intact nuclear envelope by the PLA assay (Fig. 6g and Supplementary Fig. 6c)

Reply reviewer:

This means that the IPs show pre- and post-lysis interactions. This makes me wonder what the value of the IP experiments is. In a way it shows that the M-phase specific interaction is only regulated by spatial separation. The authors should explicitly mention/explain this in the manuscript. For instance, after they conclude from the PLA that interaction is restricted to mitotic cells, they could explain that this probably means that the interactions observed by IP are mostly post-lysis.

- The authors state in line 173-176 "CIP2A efficiently accumulated in foci only in the presence of wild type TOPBP1, but not in the presence of deletion mutants, thus indicating that the efficient recruitment of both of these proteins to sites of mitotic DNA breaks is dependent on their interaction (Fig. 4b,c, Supplementary Fig. 4b)." However, the domain mapping and recruitment studies suggest that TOPBP1 774-798 Δ doesn't interact with but recruits CIP2A to DNA damage, whereas TOPBP1 813-892 Δ interacts with but doesn't recruit CIP2A to DNA damage. (Figure 4 and reviewer only figure in rebuttal). This suggest to me that interaction and recruitment are not interdependent, contrary to the author's conclusion in line 173-176.

Reply author:

We are currently unable to fully explain these observations. However, it needs to be appreciated that the interaction data have been gathered by co-immunoprecipitation of overexpressed proteins from cell extracts. This is not the same as the localization data that were collected from intact fixed cells by fluorescence microscopy. It becomes more and more clear that DNA damage foci are controlled both by stoichiometric one-to-one binding (lock and key or induced fit) associations and by highly dynamic multivalent interactions (recently reviewed by Spegg and Altmeyer, 2021, DNA Repair 106). The first one would also be observable by co-immunoprecipitation, while the second one would not, because those highly dynamic multivalent interactions may be destroyed during cell extract preparation. It is thus possible that the region between 774-798 is mainly involved in relatively "rigid" one-to-one

binding to CIP2A, while the regions between 813-891 is mostly involved in more dynamic multivalent interactions and/or TOPBP1 and CIP2A multimerization that manifests in cytologically discernible accumulation of the proteins in DNA damage foci. Both contribute to foci formation, because the 774-798 deletion mutant is clearly compromised, albeit not completely defective for accumulation at sites of DSBs.

Reply reviewer:

The explanation for the inconsistency between interaction and recruitment is not completely satisfactory, but perhaps the reader should decide whether he/she believes this conclusion. What I definitely don't like is that they claim that the interaction with the 813-892 is reduced (Figure 4), even though it is not reduced in sup fig. 4b. The authors should clarify this point and perhaps include quantifications.

- The reciprocal co-IPs for TOP1 813-892delta seem inconsistent as Figure 4a suggests this mutant is partially impaired in the interaction with CIP2A, while the reviewer only figure in the rebuttal suggest the interaction is largely intact. These are important data suggesting that TOPBP1 813-892delta may not be completely interaction-defective. These results should be presented in the supplementary information rather than the rebuttal.

Reply author:

We have included these data in the Supplementary Figures (new Supplementary Fig 4b).

Reply reviewer:

The interpretation should also be re-worded (see above).

Response to Reviewers

1. Overview

We thank the reviewers for evaluating our revised manuscript. Below, we provide a point-by-point response to all specific questions and concerns. Reviewers' comments are italicized. To distinguish our latest responses from the previous ones we have highlighted them in **bold**.

2. Point-by-point response

Reviewer #1 (Remarks to the Author): No comments for the authors.

Reviewer #2 (Remarks to the Author):

The authors have carefully considered my comments, performed several experiments to improve the manuscript, and thoroughly revised the manuscript. Nevertheless, there are a few remaining issues that I think need to be addressed before publication of the manuscript can be considered for publication.

- How can the TOPBP1-CIP2A interaction be observed in cycling, untreated cells which are mostly in cell cycle phases other than M phase (only a few percent of the cells will be in M phase), and in IR-treated cells which become arrested mainly in G2 phase? I am asking because outside M phase CIP2A has been shown to localize to the cytoplasm and not the nucleus to which TOPBP1 localizes.

Reply author:

One of the key messages of our paper is that the CIP2A-TOPBP1 interaction is controlled during the cell cycle by nuclear export of CIP2A in interphase. Since in interphase, TOPBP1 is a nuclear protein and CIP2A is mostly localized in the cytoplasm, they do not interact, because they are spatially separated by the nuclear envelope. However, upon cell extract preparation, the nuclear envelope is destroyed and CIP2A and TOPBP1 are free to interact in the cell extract. That's why we detect the interaction by co-immunoprecipitation from cell extracts of unsynchronized cell populations, even though we don't see any interaction in fixed cells with intact nuclear envelope by the PLA assay (Fig. 6g and Supplementary Fig. 6c)

Reply reviewer:

This means that the IPs show pre- and post-lysis interactions. This makes me wonder what the value of the IP experiments is. In a way it shows that the M-phase specific interaction is only regulated by spatial separation. The authors should explicitly mention/explain this in the manuscript. For instance, after they conclude from the PLA that interaction is restricted to mitotic cells, they could explain that this probably means that the interactions observed by IP are mostly post-lysis.

We agree with this Reviewer that the interaction we see in the co-immunoprecipitation experiments mostly represent post-lysis interactions. We have included textual revisions that clarify this aspect.

- The authors state in line 173-176 “CIP2A efficiently accumulated in foci only in the presence of wild type TOPBP1, but not in the presence of deletion mutants, thus indicating that the efficient recruitment of both of these proteins to sites of mitotic DNA breaks is dependent on their interaction (Fig. 4b,c, Supplementary Fig. 4b).” However, the domain mapping and recruitment studies suggest that TOPBP1 774-798delta doesn’t interact with but recruits CIP2A to DNA damage, whereas TOPBP1 813-892delta interacts with but doesn’t recruit CIP2A to DNA damage. (Figure 4 and reviewer only figure in rebuttal). This suggests to me that interaction and recruitment are not interdependent, contrary to the author’s conclusion in line 173-176.

Reply author:

We are currently unable to fully explain these observations. However, it needs to be appreciated that the interaction data have been gathered by co-immunoprecipitation of overexpressed proteins from cell extracts. This is not the same as the localization data that were collected from intact fixed cells by fluorescence microscopy. It becomes more and more clear that DNA damage foci are controlled both by stoichiometric one-to-one binding (lock and key or induced fit) associations and by highly dynamic multivalent interactions (recently reviewed by Spegg and Altmeyer, 2021, DNA Repair 106). The first one would also be observable by co-immunoprecipitation, while the second one would not, because those highly dynamic multivalent interactions may be destroyed during cell extract preparation. It is thus possible that the region between 774-798 is mainly involved in relatively “rigid” one-to-one binding to CIP2A, while the regions between 813-891 is mostly involved in more dynamic multivalent interactions and/or TOPBP1 and CIP2A multimerization that manifests in cytologically discernible accumulation of the proteins in DNA damage foci. Both contribute to foci formation, because the 774-798 deletion mutant is clearly compromised, albeit not completely defective for accumulation at sites of DSBs.

Reply reviewer:

The explanation for the inconsistency between interaction and recruitment is not completely satisfactory, but perhaps the reader should decide whether he/she believes this conclusion. What I definitely don’t like is that they claim that the interaction with the 813-892 is reduced (Figure 4), even though it is not reduced in sup fig. 4b. The authors should clarify this point and perhaps include quantifications.

We have now clarified this point by mentioning in the text that one co-IP with the 813-892 mutant showed a reduced interaction (quantified) while the reverse co-IP shown in the Supplementary Fig. 4 did not.

- The reciprocal co-IPs for TOPBP1 813-892delta seem inconsistent as Figure 4a suggests this mutant is partially impaired in the interaction with CIP2A, while the reviewer only figure in the rebuttal suggest the interaction is largely intact. These are important data suggesting

that TOPBP1 813-892delta may not be completely interaction-defective. These results should be presented in the supplementary information rather than the rebuttal.

Reply author:

We have included these data in the Supplementary Figures (new Supplementary Fig 4b).

Reply reviewer:

The interpretation should also be re-worded (see above).

As mentioned above, we have now clarified this point by specifically mentioning in the text that the reverse co-IP shown in the Supplementary Fig. 4 did not show reduced interaction.